# An improved process-oriented hydro-biogeochemical model for simulating dynamic fluxes of methane and nitrous oxide in alpine ecosystems with seasonally frozen soils

Wei Zhang[1], Zhisheng Yao[1], Siqi Li[1], Xunhua Zheng[1, 2], Han Zhang[1, 3], Lei Ma[1, 4], Kai Wang[1], Rui Wang[1], Chunyan Liu[1], Shenghui Han[1], Jia Deng[5], Yong Li[6]

[1]State Key Laboratory of Atmospheric Boundary Layer Physics and Atmospheric Chemistry, Institute of Atmospheric Physics, Chinese Academy of Sciences, Beijing 100029, P. R. China
[2]College of Earth and Planetary Science, University of Chinese Academy of Sciences, Beijing 100049, P. R. China
[3]School of Geographic and Environmental Sciences, Tianjin Normal University, Tianjin 300387, P. R. China
[4]Institute of Meteorology and Climate Research, Atmospheric Environmental Research (IMK-IFU), Karlsruhe Institute of Technology, Kreuzeckbahnstrasse 19, 82467 Garmisch-Partenkirchen, Germany
[5]Complex Systems Research Center, Institute for the Study of Earth, Oceans and Space, University of New Hampshire, 39 College Road, Durham, NH 03824, USA
[6]Key Laboratory of Agro-ecological Processes in Subtropical Region, Institute of Subtropical Agriculture, Chinese Academy of Sciences, Hunan 410125, P. R. China

*Correspondence to*: Xunhua Zheng (xunhua.zheng@post.iap.ac.cn)

**Abstract.** The hydro-biogeochemical model Catchment Nutrient Management Model - DeNitrification-DeComposition (CNMM-DNDC) was established to simultaneously quantify ecosystem productivity and losses of nitrogen and carbon at the site or catchment scale. As a process-oriented model, this model is expected to be universally applied to different climate zones, soils, land uses and field management practices. This study is one of many efforts to fulfil such an expectation, which was performed to improve the CNMM-DNDC by incorporating a physical-based soil thermal module to simulate the soil thermal regime in the presence of freeze-thaw cycles. The modified model was validated with simultaneous field observations in three typical alpine ecosystems (wetlands, meadows and forests) within a catchment located in the seasonally frozen region of the eastern Tibetan Plateau, including soil profile temperature, topsoil moisture and fluxes of methane ($CH_4$) and nitrous oxide ($N_2O$). The validation showed that the modified CNMM-DNDC was able to simulate the observed seasonal dynamics and magnitudes of above variables in the three typical alpine ecosystems, with index of agreement values of 0.91–1.00, 0.49–0.83, 0.57–0.88 and 0.26–0.47, respectively. Consistent with the emissions determined from the field observations, the simulated aggregate emissions of $CH_4$ and $N_2O$ were highest for the wetland among three alpine ecosystems, which were dominated by the $CH_4$ emissions. This study indicates the possibility for utilizing the process-oriented model CNMM-DNDC to predict hydro-biogeochemical processes, as well as related gas emissions, in seasonally frozen regions. As the original CNMM-DNDC was previously validated in some unfrozen regions, the modified CNMM-DNDC could be potentially applied to estimate the emissions of $CH_4$ and $N_2O$ from various ecosystems under different climate zones at the site or catchment scale.

## 1 Introduction

During the last 50 years, the extraordinary changes in the nitrogen and carbon cycles have occurred globally, which are essential components of ecosystems (e.g., Galloway *et al.*, 2008; Canfield *et al.*, 2010). Climate changes due to warming and human anthropogenic activities derived from food production have significantly altered the cycling of nitrogen and carbon and led to increased reactive nitrogen availability and carbon losses, which result in a series of environmental problems at the catchment, regional and even global scales (e.g., Galloway *et al.*, 2004; Galloway *et al.*, 2008; Ju *et al.*, 2009). Excessive reactive nitrogen in soils can be lost in the forms of nitrogen gases, such as nitrous oxide ($N_2O$), nitric oxide (NO) and ammonia ($NH_3$), and nitrogen pollution, such as nitrate ($NO_3^-$) and ammonium ($NH_4^+$), in water through leaching or surface runoff (e.g., Seitzinger, 2008; Collins *et al.*, 2016). In the face of increased air temperatures and intensive land use changes, especially in cold regions, the soil organic carbon stored since the Last Glacial Maximum has been lost to the atmosphere via methane ($CH_4$) and carbon dioxide ($CO_2$) (e.g., Piao *et al.*, 2009; Fenner and Freeman, 2011; Schuur *et al.*, 2015). These nitrogen and carbon losses contribute to potential global warming ($CO_2$, $CH_4$ and $N_2O$), air pollution (NO and $NH_3$) and surface/groundwater pollution ($NO_3^-$ and $NH_4^+$). Therefore, sustainable ecosystems urgently need to be established that not only focus on net primary productivity but also are friendly to the environment with the minimal hazards, including greenhouse gases, air pollution and water pollutants (e.g., Cui *et al.*, 2018; Zhang *et al.*, 2019).

The cycling of nitrogen and carbon is closely related to soil water processes (e.g., Breuer *et al.*, 2010; Vereecken *et al.*, 2016; Zhang *et al.*, 2018b). Thus, interactions among soil waters and the cycling of nitrogen and carbon govern biological productivity and environmental outcomes (e.g., Zhu *et al.*, 2018). The interactions consist of the redox potential for different transformation processes influenced by the spatiotemporal variation in soil water content and the lateral transport of water and dissolved nitrogen or carbon controlled by surface and subsurface flow (e.g., McClain *et al.*, 2003; Castellano *et al.*, 2013; Bechmann, 2014). For example, the variation in soil water content can create hot spots or moments of nitrogen and carbon losses by influencing plant nitrogen uptake, redox potential, and the transport of dissolved nitrogen and carbon (e.g., Zhu *et al.*, 2012; Keiluweit *et al.*, 2017). Therefore, a complete understanding of biogeochemical processes will inevitably involve interactions among soil water and the cycling of nitrogen and carbon (e.g., Breuer *et al.*, 2010; Vereecken *et al.*, 2016; Zhu *et al.*, 2018).

Biogeochemical models, such as DNDC, LandscapeDNDC, WNMM, MOMOS, CENTURY and DayCent, are effective tools for simulating the cycling of nitrogen and carbon and quantifying the effects of climate change and anthropogenic activities on ecosystems (e.g., Foereid *et al.*, 2007; Haas *et al.*, 2012; Li, 2007; Li *et al.*, 2007; Pansu *et al.*, 2010; Cheng *et al.*, 2014; Pansu *et al.*, 2014). In recent years, some new conceptual approaches are applied in the biogeochemical models, such as centering on the functional role of the soil microbial biomass (Pansu *et al.*, 2010; Pansu *et al.*, 2014) and detailing the lateral transport of water and nutrients (Haas *et al.*, 2012; Zhang *et al.*, 2018b). Generally, comprehensive hydrological processes, especially for the lateral transport of water and nutrients, are simplified or ignored in most models due to specific questions that must be addressed (e.g., Li, 2007; Li *et al.*, 2007; Chen *et al.*, 2008; Deng *et al.*,

2014). For the land surface or hydrological models at large scales, they are designed with explicit mechanisms of hydrology
and generally focus on vertical and lateral nutrient transport, such as nitrate loads into rivers (e.g., Liu *et al.*, 2019). However,
the simulations of nitrogen and carbon processes are usually based on empirical functions even without predicting gas loss.
Due to the various purposes of different models, coupling soil hydrological models with biogeochemical models can be an
effective strategy for integrating soil water and cycling of nitrogen and carbon to improve model performance. Thus, the
coupled model with improved performance can be applied to simultaneously predict productivity and potential negative
environmental effects (e.g., Chen *et al.*, 2008; Zhu *et al.*, 2018).
In recent years, efforts have been implemented to couple models, such as SWAT-N, LandscapeDNDC-CMF, APSIM,
SWAT-DayCent, and CNMM-DNDC (e.g., Pohlert *et al.*, 2007; Haas *et al.*, 2012; Holzworth *et al.*, 2014; Wu *et al.*, 2016;
Zhang *et al.*, 2016; Zhang *et al.*, 2018b). The models derived from SWAT were all based on semi-distributed hydrological
models using hydrologic response units and did not perform better in estimating non-point source pollution (e.g., Pohlert *et*
*al.*, 2007; Bosch *et al.*, 2011 ;Wu *et al.*, 2016). A coupler was used to couple two models for LandscapeDNDC-CMF, which
realized the simulation of horizontal movement of water and nutrients (e.g., Haas *et al.*, 2012; Klatt *et al.*, 2017; Schroeck *et*
*al.*, 2019). Compared with other models, the Catchment Nutrient Management Model - DeNitrification-DeComposition
(CNMM-DNDC), which was established by incorporating the core biogeochemical processes of DNDC into the
hydrological framework of the CNMM, was validated at a catchment with complex landscapes in the subtropical region and
showed good performance for simultaneously simulating various variables, including ecosystem productivity, hydrological
nitrogen losses and nitrate discharge in streams, and emissions of gaseous carbon and nitrogenous gases (Zhang *et al.*,
2018b). Therefore, the CNMM-DNDC has the capacity to simulate the various variables closely related to both productivity
and environmental hazards.
However, as a process-oriented hydro-biogeochemical model designed to be applicable to different climate zones, soils,
land uses and field management practices, CNMM-DNDC testing is still lacking due to limited observations for model
validation. In this study, the model was applied to a catchment in a seasonally frozen region located on the eastern Tibetan
Plateau with the land use types of alpine wetlands, meadows and forests to test its ability to simulate hydro-biogeochemical
processes. However, scientific descriptions of soil thermal dynamics due to freeze-thaw cycles are still lacking for the
CNMM-DNDC. This gap may hinder model application in seasonally frozen regions, which account for 56% of the exposed
land surface of the Northern Hemisphere (Jiang *et al.*, 2020). In addition, the soil freeze-thaw cycles that occur in these mid-
high latitude regions exert important influences on soil thermal dynamics, as well as on related hydrological processes, thus
increasing the availability of substrates and stimulating the processes of $CH_4$ and $N_2O$ production and emissions in soils (e.g.,
Song *et al.*, 2019). Therefore, we hypothesize that adding the missing scientific processes of soil thermal dynamics into the
internal model program codes can improve the performance of the CNMM-DNDC in simulating the soil thermal dynamics,
hydrological processes and $CH_4$ and $N_2O$ fluxes in seasonally frozen regions. Filling this gap is especially necessary to
broaden model applicability.
To test the above hypothesis, the catchment simulation in the Rierlangshan was conducted using a unique experimental
dataset, which was obtained by Zhang *et al.* (2018a, 2019) and Yao *et al.* (2019) for the catchment that involved three typical
alpine ecosystems, wetlands, meadows and forests, on the eastern Tibetan Plateau. The aims of this study were to (i) attempt
to address the gap in the CNMM-DNDC by improving the scientific processes of soil thermal dynamics for seasonally
frozen regions and (ii) compare the performances of the original and modified models in simulating the soil profile
temperature, topsoil moisture and $CH_4$ and $N_2O$ fluxes in three typical alpine ecosystems in the Rierlangshan catchment with
field observations. Therefore, the validated model with modifications provides a mechanism for not only interpreting
observations but also predicting the $CH_4$ and $N_2O$ fluxes in alpine ecosystems.
**2 Materials and methods**
**2.1 Model description**
**2.1.1 Overview of the CNMM-DNDC model**
The CNMM-DNDC is a process-oriented model developed for simulating hydro-biogeochemical interactions at the
catchment or site scale, and this model is designed following the basic theories of physics, chemistry, and biogeochemistry
and has the capacity to simulate the complex transport and transformation of water, nitrogen and carbon in terrestrial
ecosystems under both aerobic and anaerobic conditions. The model can be applied to simultaneously quantify ecosystem
productivity, net emissions of nitrogen and carbon gases and hydrological nitrogen losses through soil leaching and
discharge in streams from an entire catchment or individual landscape unit (Zhang *et al.*, 2018b).The model was established
to address the bottleneck issue associated with most biogeochemical models, i.e., the inability to simulate the lateral flows of
water and nutrients, by incorporating the core biogeochemical processes of DNDC (including the processes of
decomposition, nitrification, denitrification and fermentation) into the hydrological framework of the CNMM, which is fully
distributed. For the new generation of biogeochemical models, the microbial ecology was integrated into the biogeochemical
models, which represents direct microbial control over decomposition, such as MOMOS (Pansu *et al.*, 2010; Treseder *et al*.,
2011; Todd-Brown *et al*., 2012; Pansu *et al.*, 2014). The biogeochemical processes simulated by the DNDC were generally
based on first-order kinetics for decomposition and Michaelis-Menten kinetics of two substrates for nitrification and
denitrification, which only the parameterized growth and death of nitrifiers and denitrifiers were considered (Li, 2000).
However, due to the global application and validation of DNDC (e.g., Chen *et al*., 2008; Giltrap *et al*., 2010; Cui *et al*., 2014,
Zhang *et al*., 2015), the biogeochemical processes of DNDC were selected in the CNMM-DNDC despite some deficiencies
in simulating microbial biomass.
The simulated soil depth (including bedrock) is user-defined. The temporal and spatial resolutions are also user-
defined according to the driving data of climate (generally in 3 hours) and digital elevation model (DEM). The soil moisture
was calculated based on the mass balance of precipitation, irrigation, evapotranspiration, vertical flow, lateral flow and water

from a rising water table. The total water that can be infiltrated during each time step was determined by a defined maximum infiltration rate. Darcy's law was applied for predicting the vertical water flow in the soil profile. A cell-by-cell approach using a kinematic approximation was applied to route the saturated overland and subsurface flow based on DEM. The stream flow was estimated using a cascade of linear channel reservoirs (Wigmosta *et al.*, 1994). For plant growth, gross primary production was simulated using Farquhar *et al.* (1980) for $C_3$ and Collatz *et al.* (1992) for $C_4$, with annual primary productivity calculated as the residue of gross primary production and autotrophic respiration. The processes related to the production of $N_2O$ include nitrification and denitrification, which occur simultaneously at aerobic and anaerobic microsites, respectively. The concept of an "anaerobic balloon" was adopted to determine the microsites and allocate substrates for nitrification and denitrification. The sizes of the aerobic (nitrification) and anaerobic (denitrification) microsites were determined by the soil redox potential (Eh) using the Nernst equation (Li, 2007). The "hole in the pipe" concept was applied to calculate $N_2O$ production during nitrification, which is influenced by the soil moisture, temperature and pH (Li, 2016). The production of $N_2O$ during denitrification was predicted with Michaelis-Menten kinetics and Pirt functions following the reaction chain of denitrification. The predicted $CH_4$ flux was influenced by $CH_4$ production, oxidation and transportation derived from the module of fermentation in the DNDC (Li, 2007). Methane production and oxidation occurred simultaneously and were determined by the sizes of the aerobic (production) and anaerobic (oxidation) microsites, which were defined by an Eh calculator in terms of an "anaerobic balloon" ("$CH_4$ balloon") (Li, 2007). The predicted $CH_4$ production was calculated from the carbon substrates resulting from decomposed soil organic carbon (SOC) and plant root biomass with the effects of soil temperature (Li, 2000, 2016). For more details, please see Li. (2000, 2007) and Zhang *et al.* (2018b).

**2.1.2 Modifications of the CNMM-DNDC model**

In the CNMM-DNDC, the soil temperature was predicted by solving the one-dimensional heat conduction equation with the implicit method of Crank-Nicholson. However, despite the simple parameterization used for the calculation of soil heat capacity and thermal conductivity, the variations of soil temperature induced by the freeze-thaw cycles were also not considered (Table S1 of the online supplementary materials), which inevitably hindered its application in seasonally frozen regions. In this study, the CNMM-DNDC was modified by replacing the above soil thermal module by a physical based module of Northern Ecosystem Soil Temperature (Zhang *et al.*, 2003; Deng *et al.*, 2014), which can explicitly describe the energy exchange within the soil, the active layer dynamics and the soil thermal regime in the presence of freeze-thaw cycles. These modifications are indispensable for accurately simulating freeze-thaw cycles in seasonally frozen regions, which are crucial for characterizing the active layer and soil thermal dynamics, soil hydrology and nitrogen or carbon cycling in these regions. Therefore, the CNMM-DNDC with and without the above modifications are hereafter referred to as the original and modified model, respectively.

The modified thermal dynamics of the soil are calculated by the one-dimensional heat conduction equation (Eq. 1). The equation is solved numerically by converting to an explicit form (Eqs. 2–4), which is more efficient for considering the

freeze-thaw cycles (Zhang *et al.*, 2003). In the above equations, $C$ (J m$^{-3}$ °C$^{-1}$), $k$ (W m$^{-1}$ °C$^{-1}$), $T$ (°C) and $G$ (W m$^{-2}$) denote
the soil heat capacity, thermal conductivity, soil temperature and heat fluxes between layers, respectively. Both $Z$ and $D$ are
the thicknesses of the soil layer (m), $\Delta t$ is the time step of the calculation, and $l$ denotes the soil layer $l$. $S$ is the internal heat
exchange due to freezing or thawing (W m$^{-3}$) when the soil temperature is around 0 °C. The soil temperature changes
affected by freezing or thawing are determined on the basis of energy conservation, which indicate that the latent heat
released during freezing equalled the amount of heat required for the increased soil temperature and vice versa. The dynamic
soil heat capacity ($C_l$, J m$^{-3}$ °C$^{-1}$) is the weighted average of the heat capacity for five constituents, including organic matter
($C_{l,\text{OM}}$), minerals ($C_{l,\text{Min}}$), water ($C_{l,\text{Water}}$), ice ($C_{l,\text{Ice}}$) and air ($C_{l,\text{Air}}$) (Eq. 5). The values of heat capacity for organic matter,
minerals, water, ice and air were $2.5 \times 10^6$, $2.0 \times 10^6$, $4.2 \times 10^6$, $2.1 \times 10^6$ and $1.2 \times 10^3$ J m$^{-3}$ °C$^{-1}$, respectively (Huang, 2000). The
weight is the relative volumetric fraction of each constituent ($\theta_{l,\text{OM}}$, $\theta_{l,\text{Min}}$, $\theta_{l,\text{Water}}$, $\theta_{l,\text{Ice}}$, $\theta_{l,\text{Air}}$) in the soil. The dynamic
thermal conductivity ($k_l$, W m$^{-1}$ °C$^{-1}$) is calculated using the thermal conductivities of above five constituents (Eq. 6–13),
with values of 0.25 ($k_{l,\text{OM}}$), 2.9 ($k_{l,\text{Min}}$), 0.57 ($k_{l,\text{Water}}$), 2.2 ($k_{l,\text{Ice}}$) and 0.025 ($k_{l,\text{Air}}$) W m$^{-1}$ °C$^{-1}$ for organic matter, minerals,
water, ice and air, respectively (Johansen, 1975). ST$_l$ denotes the soil temperature of layer $l$ (°C). The upper and lower
boundary conditions of the thermal dynamics were determined by the surface energy balance and the defined geothermal
heat flux at a soil depth of 35 m.

$$C\frac{\partial T}{\partial t} = \frac{\partial}{\partial Z}(k\frac{\partial T}{\partial Z}) + S \tag{1}$$

$$C_l\frac{\Delta T_l}{\Delta t} = \frac{G_{l-1,\,l} - G_{l,\,l+1}}{D_l} + S_l \tag{2}$$

$$G_{l-1,\,l} = \frac{(0.5k_l + 0.5k_{l-1})(T_{l-1} - T_l)}{0.5D_{l-1} + 0.5D_l} \tag{3}$$

$$G_{l,\,l+1} = \frac{(0.5k_l + 0.5k_{l+1})(T_l - T_{l-1})}{0.5D_l + 0.5D_{l+1}} \tag{4}$$

$$C_l = C_{l,\text{OM}}\theta_{l,\text{OM}} + C_{l,\text{Min}}\theta_{l,\text{Min}} + C_{l,\text{Water}}\theta_{l,\text{Water}} + C_{l,\text{Ice}}\theta_{l,\text{Ice}} + C_{l,\text{Air}}\theta_{l,\text{Air}} \tag{5}$$

$$k_l = \frac{\theta_{l,\text{Water}}k_{l,\text{Water}} + F_{l,\text{Air}}\theta_{l,\text{Air}}k_{l,\text{Air\_adj}} + F_{l,\text{OM+Min}}(\theta_{l,\text{OM}} + \theta_{l,\text{Min}})k_{l,\text{OM+Min}} + F_{l,\text{Ice}}\theta_{l,\text{Ice}}k_{l,\text{Ice}}}{\theta_{l,\text{Water}} + F_{l,\text{Air}}\theta_{l,\text{Air}} + F_{l,\text{OM+Min}}(\theta_{l,\text{OM}} + \theta_{l,\text{Min}}) + F_{l,\text{Ice}}\theta_{l,\text{Ice}}} \tag{6}$$

$$k_{l,\text{Air\_adj}} = \begin{cases} k_{l,\text{Air}} + 0.0238e^{0.0536\text{ST}_l} & (\theta_{l,\text{Water}} > 0.09) \\ 0.418 \times (0.0615 + 1.96\theta_{l,\text{Water}}) & (\theta_{l,\text{Water}} \geq 0.09) \end{cases} \tag{7}$$

$$\text{g\_a} = \begin{cases} 0.333 - \dfrac{0.298\theta_{l,\text{Air}}}{1 - \theta_{l,\text{OM}} - \theta_{l,\text{Min}}} & (\theta_{l,\text{Water}} > 0.09) \\ 0.013 + 0.944\theta_{l,\text{Water}} & (\theta_{l,\text{Water}} \geq 0.09) \end{cases} \tag{8}$$

$$\text{g\_c} = 1.0 - 2.0\text{g\_a} \tag{9}$$

$$k_{l,\text{OM+Min}} = k_{l,\text{OM}}^{\frac{\theta_{l,\text{OM}}}{\theta_{l,\text{OM}} + \theta_{l,\text{Min}}}} + k_{l,\text{Min}}^{\frac{\theta_{l,\text{Min}}}{\theta_{l,\text{OM}} + \theta_{l,\text{Min}}}} \tag{10}$$

$$F_{l,\text{Air}} = 0.333\left(\frac{2.0}{1.0 + g\_a\left(\frac{k_{l,\text{Air\_adj}}}{k_{l,\text{Water}}} - 1.0\right)} + \frac{1.0}{1.0 + g\_c\left(\frac{k_{l,\text{Air\_adj}}}{k_{l,\text{Water}}} - 1.0\right)}\right) \tag{11}$$

$$F_{l,\text{OM+Min}} = 0.333\left(\frac{2.0}{1.0 + 0.125\left(\frac{k_{l,\text{OM+Min}}}{k_{l,\text{Water}}} - 1.0\right)} + \frac{1.0}{1.0 + 0.75\left(\frac{k_{l,\text{OM+Min}}}{k_{l,\text{Water}}} - 1.0\right)}\right) \tag{12}$$

$$F_{l,\text{Ice}} = 0.333\left(\frac{2.0}{1.0 + 0.125\left(\frac{k_{l,\text{Ice}}}{k_{l,\text{Water}}} - 1.0\right)} + \frac{1.0}{1.0 + 0.75\left(\frac{k_{l,\text{Ice}}}{k_{l,\text{Water}}} - 1.0\right)}\right) \tag{13}$$


Compared to the original thermal module, the internal heat exchange due to freezing or thawing ($S$) was included with
improved algorithm for thermal conductivity ($k$). In addition, the one-dimensional heat conduction equation (Eq. 1) was
solved by converting it to an explicit form in the modified model (Eqs. 2–4), while was solved with the implicit method in
the original models (Table S1). The modified CNMM-DNDC was able to simulate the thermal dynamics in seasonally
frozen regions as well as their impacts on biogeochemical processes, such as the emissions of nitrogen and carbon gases.
**2.2 Catchment and field descriptions**
The study area is the Rierlangshan catchment (34°02′N, 102°43′E) on the eastern Tibetan Plateau with an area of 189
ha (Yao *et al.*, 2019). This catchment is located in the source region of the Pai-Lung River, which is a sub-branch of the
upper Yangtze River (Zhang *et al.*, 2018a; 2019). This region is subject to a cold humid continental monsoon climate, and it
had an annual mean air temperature of $1.6 \pm 0.7$ ℃ and average annual precipitation of $649 \pm 94$ mm in 1980–2012 as
observed at the Zoige Meteorological Station (~80 km south of the catchment) (Ma *et al.*, 2018). The catchment consists of
alpine wetlands, meadows and forests (Yao *et al.*, 2019). The alpine wetlands in the catchment are part of the Zoige wetland
and are degraded due to anthropogenic drainage and climate warming (Dong *et al.*, 2010; Li *et al.*, 2014). Degraded alpine
wetlands are commonly distributed throughout the Zoige wetland, and nearly 83% of the permanently inundated wetlands
have been converted into "wet grassland" (Xiang *et al.*, 2009; Li *et al.*, 2014).
$CH_4$ and $N_2O$ fluxes were manually measured once or twice per week using the gas chromatograph-based static opaque
chamber method (Zhang *et al.*, 2018a) at three sites in alpine wetlands (34°02′6.53″N, 102°43′29.66″E, 3304 m a.s.l.),
meadows (34°02′01″N, 102°43′28″E, 3326 m a.s.l.) and forests (34°01′47.13″N, 102°44′0.87″E, 3415 m a.s.l.) in the
Rierlangshan catchment from 2013 to 2015 (Zhang *et al.*, 2018a; Yao *et al.*, 2019; Zhang *et al.*, 2019) (Fig. S1). Each
chamber was wrapped with a layer of styrofoam and aluminium foil to mitigate temperature increases inside the enclosures
due to the heating of solar radiation. The alpine wetland site is located at a slope base with a slope of 2 °. The wetland has
suffered from anthropogenic drainage and climate warming, and thus degraded to be seasonally inundated. The alpine
meadow site neighbours the alpine wetland site, which is located on the north-facing slope with gradient of 11 °. In addition,
soil temperatures at different depths and topsoil moisture in the alpine wetlands, meadows and forests were observed daily
and twice per week, respectively. The details regarding the available field observations of the $CH_4$ and $N_2O$ fluxes and the
relevant auxiliary variables are described in Table S2.

**2.3 Model simulation**

The modified CNMM-DNDC was applied in the Rierlangshan catchment with the three alpine ecosystems: wetlands,

meadows and forests. The dataset required for the catchment simulation included (1) a digital elevation model (DEM) with a
resolution of $30 \times 30$ m$^2$ from the geospatial data cloud (Fig. S1; http://www.gscloud.cn/); (2) a map of alpine ecosystems,
including wetlands, meadows and forests; (3) a climate dataset of 3-hour weather data (air temperature, precipitation, wind
speed, solar radiation, longwave radiation, and humidity), which were obtained from the meteorological station in the target
catchment for the years with field observations (2013.11–2015.10) and were adapted from the daily data at the Zoige
Meteorological Station (provided by the National Meteorological Information Center: http://data.cma.cn/; last access: 10[th]
June, 2020) for other years; (4) a soil properties dataset of the observed clay fraction, organic matter content, total nitrogen,
pH and bulk density of the three alpine ecosystems in 1 m soil profile (Ma *et al.*, 2018; Zhang *et al.*, 2018a; Yao *et al.*, 2019;
Zhang *et al.*, 2019; Table S3); and (5) a management practices dataset including grazing time and intensity for the alpine
wetlands and meadows (Table S3). In addition, other required soil inputs of field capacity, wilting point and saturated
hydrological conductivity were calculated by pedo-transfer functions (Li *et al.*, 2019; Table S4). The simulated soil depth
was defined as 35 m due to the lower boundary conditions of the thermal dynamics, which was set as the geothermal heat
flux at a soil depth of 35 m. The simulated soil profile (0−35 m depth) was divided into 23 layers, including the soil (0−1.5
m) and bedrock (1.5−35 m). The layer thicknesses of the soil (0−1.5 m) were 1, 5 10 and 50 cm for the depth of 0–10, 10–20,
20–100 and 100–150 cm, respectively. The layer thicknesses of the bedrock (1.5−35 m) were 3.5 and 31m for the depth of
1.5−4.0 and 4.0−35 m, respectively. The geothermal heat flux in the catchment was estimated at 0.053 W m$^{-2}$ (Pollack and
Chapman, 1977). For the target catchment, the soil water dynamics of the alpine ecosystems were determined by the
precipitation, evapotranspiration, infiltration, penetration and lateral flow. Using the database, a catchment simulation of
hydro-biogeochemical processes was performed with spatial and temporal resolutions of $30 \times 30$ m$^2$ and 3 hours, respectively,
by the modified CNMM-DNDC from 2012 to 2015, which could reflect the influences of hydrological processes on soil
water dynamics. Thus, the soil water dynamics of the seasonally inundated wetlands were determined by the hydrological
processes without any artificial disturbances in the catchment simulation.

**2.4 Statistics and analysis**

The statistical criteria applied for evaluating the model performance in this study included (i) the index of agreement

(IA), (ii) the Nash–Sutcliffe efficiency (NSE), and (iii) the determination coefficient ($R^2$) and slope of the zero-intercept
univariate linear regression (ZIR) of the observations against the simulations (e.g., Nash and Sutcliffe, 1970; Willmott and
Matsuura, 2005; Moriasi *et al.*, 2007; Congreves *et al.*, 2016; Jiang, 2010; Dubache *et al.*, 2019). A value of IA (0–1) closer
to 1 showed a better simulation. An NSE value (ranging from minus infinity to 1) closer to 1 was better. Better model
performance was indicated by a slope and an $R^2$ value that were both closer to 1 in a significant ZIR. For more details on
these criteria, refer to the Eqs. S1−4 in Table S5. In addition, the SPSS Statistics Client 19.0 (SPSS Inc., Chicago, USA) and
Origin 8.0 (OriginLab, Northampton, MA, USA) software packages were applied for the statistical analysis and graphical
comparison.

## 241  3 Results

### 242  3.1 Model validation

### 243  3.1.1 Soil temperature and moisture

The profile soil temperatures were observed for alpine wetland and meadow, but only topsoil temperature was
observed for the alpine forest, which could be used for model validation. The simulated soil temperatures of the three typical
alpine ecosystems were significantly improved by including the scientific processes of soil thermal dynamics suitable for
seasonally frozen regions (Figs. 1 and S2). The simulated seasonal dynamics and magnitudes were consistent with those
from the field observations for various soil depths, with IA, NSE, and ZIR slopes and $R^2$ values of 0.91−1.00, 0.68−0.99,
0.83−1.09 and 0.73−1.00 for the three alpine ecosystems, respectively (Table 1). For the observed alpine wetlands and
meadows, the simulation showed that the freezing of soil started in early November and continued to the end of April in the
next year. The frozen depth reached a maximum in the middle of February. However, the simulated maximum frozen depths
for the observed alpine meadows (0.69−0.74 m) were approximately double those for the alpine wetlands (0.30−0.39 m) (Fig.
S3).
For the soil moisture, only topsoil moisture was observed in the three alpine ecosystems, which could be applied for
model validation. The simulated topsoil moisture dynamics were comparable to those from the field observations, with IA
and NSE values of 0.49−0.83 and -0.80−0.32 for the three alpine ecosystems, respectively (Fig. 2 and Table 1). In
comparison to the other alpine ecosystems, the alpine wetlands had higher soil moisture, which ranged from 0.41 to 0.98 and
from 0.38 to 0.93 for the observations and simulations in the water-filled pore space (WFPS), respectively. The soil moisture
values of the alpine meadows and forests were highly variable and depended on the variation trend in precipitation for both
observations and simulations. However, an underestimation of soil moisture in the winter period occurred for both alpine
meadows and forests due to a possible overestimation of evapotranspiration. The performances of the modified model in
simulating the soil profile temperature and topsoil moisture indicate that the modified CNMM-DNDC can generally predict
the soil thermal and topsoil moisture dynamics in the three alpine ecosystems, which is crucial for correctly simulating soil
hydrology, plant growth and biogeochemical processes.

### 3.1.2 Methane fluxes

The daily observed $CH_4$ emissions from the alpine wetlands were highly variable and showed a clear seasonal cycle, with intensive $CH_4$ emissions from May to November and weak emissions in other periods (Fig. 3a). The observed alpine meadows and forests functioned exclusively as sinks of atmospheric $CH_4$ with higher rates of uptake during the growing season and lower uptake rates in the dormant season (Figs. 3b−c). The original model significantly overestimated $CH_4$ emissions from the alpine wetlands. The modified CNMM-DNDC accurately identified the functions of the sources or sinks in the three alpine ecosystems and generally captured the magnitude and seasonal characteristics of the daily $CH_4$ fluxes, with an IA of 0.57−0.88 for the three alpine ecosystems (Figs. 3a−c and Table 1). However, the $CH_4$ uptake rates during the dormant season were obviously underestimated by the modified model at both sites, especially at the alpine forest site, which was responsible for the underestimation of cumulative $CH_4$ uptake. The observed cumulative $CH_4$ emissions ranged from -2.60 to 33.5 kg C ha$^{-1}$ yr$^{-1}$ and the modelled values ranged from -1.90 to 31.0 kg C ha$^{-1}$ yr$^{-1}$ (Fig. 4a). For the catchment simulation, the simulated annual $CH_4$ emissions ranged from -2.35 to 73.0 kg C ha$^{-1}$ yr$^{-1}$ from November 2013 to November 2014 (Fig. S4a). These results indicate that the modified CNMM-DNDC well simulated the $CH_4$ fluxes of the three typical alpine ecosystems.

### 3.1.3 Nitrous oxide fluxes

The daily observed $N_2O$ emissions from the alpine wetlands were higher than those from the alpine meadows but lower than those from the alpine forests (Figs. 3d−f). Similar seasonal patterns of $N_2O$ fluxes were observed for the three alpine ecosystems with intensive emissions in the growing season. The $N_2O$ emission peak during the dormant season was observed in the alpine meadows, which was the major contributor to annual emissions. The modified CNMM-DNDC generally captured the seasonal dynamics of daily $N_2O$ fluxes with an IA of 0.26−0.47 for the three alpine ecosystems (Figs. 3d−f and Table 1), but the $N_2O$ emissions from the alpine wetlands were significantly overestimated by the original model. For the modified model, the simulated $N_2O$ emissions from the alpine wetlands and forests showed obvious seasonal patterns with higher emissions during the growing season, but no abrupt emission peak was captured at the end of the growing season for the alpine wetlands. In addition, compared with the original model, the modified model captured the peak emissions that occurred during the freeze-thaw period from the alpine meadows due to the death of microbes, but the dynamics of the peak emissions were not well simulated. The observed cumulative $N_2O$ emissions ranged from 0.14 to 0.58 kg N ha$^{-1}$ yr$^{-1}$ and the modelled values ranged from 0.12 to 0.32 kg N ha$^{-1}$ yr$^{-1}$ (Fig. 4b). For the catchment simulation, the simulated annual $N_2O$ emissions ranged from 0.01 to 0.74 kg N ha$^{-1}$ yr$^{-1}$ from November 2013 to November 2014 (Fig. S4b). These results indicate that the modified CNMM-DNDC has the potential to estimate $N_2O$ emissions in seasonally frozen regions.

## 3.2 Annual aggregate emissions of CH$_4$ and N$_2$O

Annual aggregate emissions of CH$_4$ and N$_2$O in carbon dioxide (CO$_2$) equivalents were calculated for the three alpine ecosystems from November 2013 to November 2014 for alpine wetlands and meadow and from April 2014 to April 2015 for alpine forests, and the global warming potentials were 34 for CH$_4$ and 298 for N$_2$O on a 100-year time horizon (IPCC, 2013). The simulated aggregate emissions by the modified model were 1.5, 0.015, and 0.061 Mg CO$_2$eq ha$^{-1}$ yr$^{-1}$ for the observed alpine wetlands, meadows and forests, respectively, which were consistent with those from the field observations (1.6, 0.014, and 0.15 Mg CO$_2$eq ha$^{-1}$ yr$^{-1}$ for the alpine wetlands, meadows and forests, respectively) (Fig. 4c). However, the original model significantly overestimated the aggregate emissions due to the high predicted CH$_4$ and N$_2$O emissions. In comparison, the observed seasonally inundated wetlands functioned as the sources of aggregate emissions of CH$_4$ and N$_2$O, but the aggregate emissions from adjacent wet alpine meadows were much lower.

## 4 Discussions

### 4.1 Model performance in simulating thermal dynamics

The soil freeze-thaw cycles in seasonally frozen regions determine the soil profile temperature and hydrological processes, which are key factors that regulate the cycling of nitrogen and carbon (e.g., Zhang *et al.*, 2015; Hugelius *et al.*, 2020). Therefore, improving the scientific processes of soil thermal dynamics in the presence of active layer dynamics is essential for applying the CNMM-DNDC to simulate the biogeochemical processes in seasonally frozen regions, which are sensitive and vulnerable to climate change and human activities (Hatano, 2019; Hugelius *et al.*, 2020; Jiang *et al.*, 2020). The original model adopted a relatively simple module to calculate thermal transportation within the soil profile and did not consider the effects of freeze-thaw cycles on soil temperature and moisture. The newly incorporated module was based on explicit energy conservation and exchange in the soil profile and successfully captured the variations in soil temperature and topsoil moisture for the three alpine ecosystems during the freeze-thaw period. The simulated lower soil frozen depth for the observed alpine wetland was primarily attributed to the higher soil profile moisture level, as the thermal conductivity and heat capacity for water-filled pores were higher than those for air-filled pores. In order to quantify the impacts of climate change on the cycling of carbon and water on the regional and global scales, several large scale ecosystem models or macroscale hydrological models, such as Terrestrial Ecosystem Model, Lund-Potsdam-Jena dynamic global vegetation model and Variable Infiltration Capacity model, have been enhanced to simulate the soil thermal dynamics at northern high latitude (e.g., Wania *et al.*, 2009; Zhuang *et al.*, 2001; Cuo *et al.*, 2015; Jiang *et al.*, 2020). In addition, the soil thermal modules were also improved in some biogeochemical models, such as DNDC and Mobile-DNDC, to evaluate the influences of climate warming on the biogeochemical processes in high latitude regions (e.g., Zhang *et al.*, 2003; de Bruijn *et al.*, 2009; Wolf *et al.*, 2011; Zhang *et al.*, 2012; Deng *et al.*, 2014). Compared with the simulated soil profile temperatures by above models at different scales, the simulations in this study by the modified CNMM-DNDC were equally well, especially for

deeper soil layers (e.g., Wania *et al.*, 2009). For the validated topsoil moisture in this study, the modified model generally
captured the variation trends, which were comparable with the performances of other models (e.g., de Bruijn *et al.*, 2009;
Wolf *et al.*, 2011; Cuo *et al.*, 2015). However, compared with the studies focused on simulating soil moisture (e.g., Ford *et*
*al.*, 2014), further improvements are still required to improve the model performance in simulating the soil moisture. These
results indicate the efficiency of the incorporated module in simulating soil thermal and topsoil moisture dynamics in
seasonally frozen regions.

## 4.2 Model performance in simulating $CH_4$ fluxes

Compared with the annually inundated wetlands, the seasonally inundated wetlands had relatively low observed and

simulated $CH_4$ emissions due to the significant influences of the water table level on $CH_4$ emissions (e.g., Hatano, 2019;
Zhang *et al.*, 2019). The $CH_4$ emissions simulated by the CNMM-DNDC were determined by the processes of production,
oxidation and transpiration. The unsaturated soil with moisture levels ranging from 0.41 to 0.98 WFPS resulted in a small
$CH_4$ balloon and thus reduced $CH_4$ production. At the same time, relatively dry conditions caused the upper soil layer to act
as an efficient oxidative methanotrophic barrier for the diffusion of $CH_4$ from the subsoil and thus decreased $CH_4$ emissions
(Kandel *et al.*, 2018; Tan *et al.*, 2020). In addition, the highly fluctuating $CH_4$ emissions simulated by the modified model
were also attributed to the high dependency of $CH_4$ production on soil moisture, which controlled the size of the $CH_4$ balloon.
Theoretically, the $CH_4$ emissions simulated by the original model should not be higher than those simulated by the modified
model due to the lower predicted soil moisture level. The overestimated $CH_4$ emissions simulated by the original model were
mainly attributed to the overestimated soil temperature due to their influences on mineralized substrates for $CH_4$ production,
as well as the processes of $CH_4$ production. This result implies that global warming may trigger intensive $CH_4$ emissions
from degraded wetlands, which could partly serve as a trade-off for the decreased $CH_4$ emissions due to the lower water table
level in degraded wetlands (e.g., Gong *et al.*, 2020). For the studies focused on simulating $CH_4$ emissions from wetlands by
the large-scale ecosystem models, the model validation with field observation is difficult due to coarse spatial resolution (e.g.,
Zhuang *et al.*, 2004). For the biogeochemical model, such as DNDC, the dynamics of $CH_4$ emissions from wetland and
peatland in the northern permafrost regions were well simulated (Zhang *et al.*, 2012; Deng *et al.*, 2014), which showed
consistent seasonal variations and magnitudes as those in this study. Both observations and simulations showed that the $CH_4$
uptake in alpine forests was higher than that in alpine meadows, which was mainly attributed to the high SOC content of the
alpine forests in the simulation. Methane uptake by upland soils is a biological process governed by the availability of $CH_4$
and oxygen as well as the activity and quantity of methanotrophic bacteria in soils (e.g., Liu *et al.*, 2007; Zhang *et al.*, 2014).
In the model, the simulated $CH_4$ uptake was positively related to the SOC content, which is closely related to the population
size of methanotrophic bacteria. Thus, the SOC content primarily contributed to the differences in $CH_4$ uptake from alpine
meadows and forests, as the values for forests were more than twice of those for meadows (Table S3). As the simulated
dynamic characteristics of $CH_4$ uptake were primarily regulated by soil temperature and moisture, the inhibitory effects of
low soil temperature ($< 0.0$ ℃) on $CH_4$ uptake rates resulted in obvious underestimations in the dormant season for both
alpine meadows and forests. Therefore, an improved parameterization for simulating $CH_4$ uptake under low soil temperatures
is required for the model to better capture the dynamics of $CH_4$ uptake in the dormant season.

## 4.3 Model performance in simulating $N_2O$ fluxes

In comparison, the $N_2O$ emissions from the alpine wetlands and forests were higher than those from the alpine
meadows for both the observations and simulations due to the high SOC content and nitrogen availability. Natural wetlands
are large carbon reserves and play a crucial role in mitigating global warming (e.g., Deng *et al.*, 2014; Kang *et al.*, 2020; Tan
*et al.*, 2020). The intentional drainage of annually inundated wetlands alters not only the water regime but also nutrient
availability (e.g., Hoffmann *et al.*, 2016). The simulated relatively low soil moisture for the alpine wetlands stimulated the
decomposition of SOC and nitrogen (or peat oxidation) under aerobic conditions, thus improving nitrogen mineralization for
nitrification and denitrification and enhancing $N_2O$ emissions (e.g., Tan *et al.*, 2020; Zhang *et al.*, 2020). The intensive $N_2O$
emissions simulated by the original model resulted from the overestimated soil temperature for the alpine wetlands. Firstly,
as the presence of ice could impede the water movement, the water lateral flows were promoted by the original model due to
the neglecting of freeze-thaw cycles. These further resulted in the lower simulated soil moisture as compared with the
modified model (Fig. S5), which provided favorable oxygen conditions for $N_2O$ production. Meanwhile, the simulated high
soil moisture by the modified model provided feasible anaerobic conditions for thoroughly denitrification. Secondly, higher
simulated soil temperature by the original model also facilitated the mineralization, which provided more available mineral
nitrogen. Field studies showed that high SOC concentrations could stimulate the processes of mineralization and nitrification
in the forests (e.g., Li *et al.*, 2005; Yao *et al.*, 2019). The model input of soil organic matter measured in the observed alpine
forests was more than twice that in the observed alpine meadows (Table S3). Thus, the high SOC content at the alpine forest
site provided more available nitrogen through mineralization and thus stimulated the nitrification processes in the simulation.
Furthermore, the seasonal grazing that occurred in the alpine meadows resulted in constant loss of available nitrogen and
thus hindered the $N_2O$ emissions from the biological processes in the simulation. Field observations showed that the soil
freeze-thaw cycles occurred in seasonally frozen regions not only increased the availability of nitrogen and carbon substrates
by disrupting of soil aggregates but also affected the structure, population and activity of the microbes, and thus influencing
the emissions of $N_2O$ (e.g., Song *et al.*, 2019). de Bruijn *et al.* (2009) have explored the combined mechanisms for
simulating freeze–thaw related $N_2O$ emissions, which were the promoted anaerobiosis and denitrification due to reduced gas
diffusion derived from soil frost and snow cover, and the stimulated microbial growth due to easy decomposable organic
carbon and nitrogen derived from the dead microbes during freeze-thaw cycles. Wolf *et al.* (2011) introduced an impedance
factor to parameterize the reduced water flow between layers in the presence of ice, which could captured the freeze–thaw
related $N_2O$ emissions for ungrazed steppe. In the CNMM-DNDC, threshold values of soil temperature were set to trigger
the death of microbes during the freezing period and stimulate the production of NO, $N_2O$ and $N_2$ using substrates derived
from the dead microbes during the thawing period, which was similar to one of the mechanisms explored by de Bruijn *et al.*
(2009). However, compared with the simulated freeze–thaw related $N_2O$ emissions by other studies, the simulated dynamics

of peak emissions due to freeze-thaw cycles in this study were inconsistent with those from the field observations. Thus, improvements are required to incorporate some other effective mechanisms to better capture the dynamic characteristics. The peak emissions during the freeze-thaw period were not captured by the original model due to the significantly overestimated soil temperature. The low evaluation statistics for the daily fluxes, especially for the alpine forests, were also attributed to the underestimation of background emissions, which resulted from both measurement errors due to low fluxes around detection limits ($\pm 0.41$ g N ha$^{-1}$ d$^{-1}$) and model deficiencies in the simulation of tight nitrogen cycling in natural ecosystems.

Compared with the empirical model, one key advantage of the process-oriented models is that the models are independent of the local parameterization (Zhang *et al.*, 2015). In this study, default internal parameter combinations of biogeochemical processes were used for the original and modified models, which have been applied in the catchment simulation in the subtropical region (Zhang *et al.*, 2018b), due to the limited field observations (only one year) for both calibration and validation. The biogeochemical processes were predicted by the first-order and Michaelis-Menten kinetics in the CNMM-DNDC based on some defined parameters of flow fractionation. For instance, there are 17 parameters related with $N_2O$ emission in the module of denitrification (Table S6), which would inevitably increase the uncertainty of simulation. Houska *et al.* (2017) found that hydro-biogeochemical models can be right for the wrong reasons, such as matching greenhouse gas emissions while failing to simulate soil moisture, which emphasized the importance for simultaneous validations of multi-variables. Thus, simultaneous validations of $CH_4$ and $N_2O$ fluxes, as well as soil environment variables, were necessary for comprehensive evaluation of the model performance. In addition, the microbial ecology was recently recommended to be integrated into the biogeochemical model using a smaller number of well-defined kinetic parameters, such as MOMOS (Pansu *et al.*, 2010; Treseder *et al.*, 2011). Therefore, direct control of microbial on biogeochemical processes, such as the stoichiometry of decomposer, is required to be included in the CNMM-DNDC in near future. The model performances of simulating various variables for three typical alpine ecosystems in the Rierlangshan catchment imply that the modified CNMM-DNDC can be applied to predict the thermal dynamics and fluxes of $CH_4$ and $N_2O$ from alpine ecosystems in seasonally frozen regions.

## 5 Conclusions

To apply the process-oriented hydro-biogeochemical model Catchment Nutrient Management Model - DeNitrification-DeComposition (CNMM-DNDC) in seasonally frozen regions, an improved module of soil thermal dynamics for describing the soil thermal regime in the presence of freeze-thaw cycles was incorporated in this study. Using the unique experimental dataset obtained in the Rierlangshan catchment with the typical alpine wetland, meadow and forest ecosystems, the modified model was evaluated for simulating soil thermal dynamics (soil profile temperature), topsoil moisture and fluxes of methane ($CH_4$) and nitrous oxide ($N_2O$) in seasonally frozen regions of the Tibetan Plateau. The modified CNMM-DNDC could generally capture the seasonal dynamics and magnitudes of profile soil temperature, topsoil moisture and fluxes of $CH_4$ and $N_2O$ in seasonally frozen regions. Both the observed and simulated $CH_4$ and $N_2O$ fluxes from three alpine ecosystems

indicate that the aggregate emissions of $CH_4$ and $N_2O$ were highest for the wetland among three alpine ecosystems. The intensive aggregate emissions of $CH_4$ and $N_2O$ were regulated by the high soil moisture, which was primarily determined by the $CH_4$ emissions. This study implies that a hydro-biogeochemical model, such as the modified CNMM-DNDC, are able to predict soil thermal dynamics, topsoil moisture and fluxes of $CH_4$ and $N_2O$ in seasonally frozen regions with an improved physical-based soil thermal module.

**Data availability**

The model, input and output databases can be obtained from the first author and all the observed data sets used in this study can be available from the co-authors.

**Author contribution**

Zheng, X. and Zhang, W. contributed to developing the idea and enhancing the science of this study. Zhang, W. improved the scientific processes of the model, implemented the model simulations and prepared the manuscript with contributions from all co-authors. Li, S. improved the model structure for standard input. Yao, Z., Zhang, H., Ma, L., Wang, K., Wang, R. and Liu, C. designed and carried out the field experiments. Han, S. collected and established the input database for modelling. Deng, J and Li, Y contributed to the modification of the model and the improvement of the manuscript.

**Competing interests**

The authors declare that they have no conflict of interest.

**Acknowledgement**

This study was jointly supported by the National Key R&D Program of China (2016YFA0602303), the Chinese Academy of Sciences (ZDBS-LY-DQC007), the National Key Scientific and Technological Infrastructure project "Earth System Science Numerical Simulator Facility" (EarthLab) and the National Natural Science Foundation of China (41603075, 41861134029).

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

Table 1 Statistics of the validated variables by the modified CNMM-DNDC for three typical alpine ecosystems.

| Item | Ecosystem | $n$[a] | DR[b] | IA | | NSE | | ZIR-slope | | ZIR-$R^{2}$[e] | | ZIR-P | |
|---|---|---|---|---|---|---|---|---|---|---|---|---|---|
| | | | | O[c] | M[d] | O | M | O | M | O | M | O | M |
| Soil temperature | | | | | | | | | | | | | |
| 5 cm | Meadow | 500 | Daily | 0.90 | 0.96 | 0.82 | 0.95 | 0.84 | 1.09 | 0.89 | 0.96 | < 0.001 | < 0.001 |
| | Forest | 48 | Daily | 0.85 | 0.91 | 0.37 | 0.68 | 0.64 | 0.83 | 0.68 | 0.73 | < 0.001 | < 0.001 |
| 10 cm | Wetland | 366 | Daily | 0.90 | 0.98 | 0.57 | 0.92 | 0.72 | 1.07 | 0.81 | 0.93 | < 0.001 | < 0.001 |
| | Meadow | 500 | Daily | 0.93 | 0.99 | 0.71 | 0.95 | 0.80 | 1.08 | 0.85 | 0.96 | < 0.001 | < 0.001 |
| 20 cm | Wetland | 366 | Daily | 0.82 | 0.99 | 0.18 | 0.96 | 0.64 | 1.05 | 0.66 | 0.97 | < 0.001 | < 0.001 |
| | Meadow | 500 | Daily | 0.87 | 0.99 | 0.48 | 0.97 | 0.74 | 1.06 | 0.76 | 0.98 | < 0.001 | < 0.001 |
| 50 cm | Wetland | 366 | Daily | 0.66 | 0.99 | -1.01 | 0.97 | 0.51 | 1.05 | 0.43 | 0.97 | < 0.001 | < 0.001 |
| | Meadow | 401 | Daily | 0.70 | 1.00 | -0.48 | 0.99 | 0.58 | 1.06 | 0.53 | 1.00 | < 0.001 | < 0.001 |
| 70 cm | Wetland | 366 | Daily | 0.58 | 0.98 | -2.23 | 0.93 | 0.47 | 1.05 | 0.38 | 0.93 | < 0.001 | < 0.001 |
| | Meadow | 401 | Daily | 0.64 | 1.00 | -1.19 | 0.99 | 0.54 | 1.03 | 0.49 | 1.00 | < 0.001 | < 0.001 |
| 90 cm | Wetland | 366 | Daily | 0.52 | 0.98 | -4.07 | 0.90 | 0.44 | 1.03 | 0.36 | 0.90 | < 0.001 | < 0.001 |
| Soil moisture | Wetland | 74 | Daily | 0.63 | 0.83 | -1.65 | 0.20 | 1.31 | 1.13 | – | 0.60 | – | < 0.001 |
| | Meadow | 128 | Daily | 0.78 | 0.78 | 0.28 | 0.32 | 0.96 | 0.93 | 0.30 | 0.41 | < 0.001 | < 0.001 |
| | Forest | 40 | Daily | 0.48 | 0.49 | -1.04 | -0.80 | 1.21 | 1.19 | – | – | – | – |
| Daily CH$_4$ flux | Wetland | 180 | Daily | 0.37 | 0.74 | -11.1 | -0.73 | 0.46 | 0.87 | – | – | – | – |
| | Meadow | 168 | Daily | 0.87 | 0.88 | 0.42 | 0.38 | 1.09 | 0.94 | 0.44 | 0.39 | < 0.001 | < 0.001 |
| | Forest | 49 | Daily | 0.59 | 0.57 | -2.79 | -3.39 | 0.92 | 0.79 | – | – | – | – |
| DailyN$_2$O flux | Wetland | 180 | Daily | 0.01 | 0.26 | -323 | -0.07 | 0.01 | 0.59 | – | – | – | – |
| | Meadow | 168 | Daily | 0.23 | 0.44 | -0.16 | -1.76 | 0.99 | 0.35 | – | – | – | – |
| | Forest | 58 | Daily | 0.47 | 0.47 | -1.85 | -1.64 | 0.44 | 0.47 | – | – | – | – |

[a] $n$ indicates the number of the observations. [b] DR denotes the time resolution of the observed data. [c] O indicates the simulations by the original
model. [d] M indicates the simulations by the modified model. [e] "–" indicated no value due to the sum of regression square are larger than the sum
of the total square for the regression). IA, NSE, ZIR-slope, ZIR-$R^2$ and ZIR-P indicate the index of agreement, Nash–Sutcliffe efficiency,
determination coefficient and slope of the zero-intercept univariate linear regression (ZIR) of the observations against the simulations, as well as
the significance level of the ZIR.

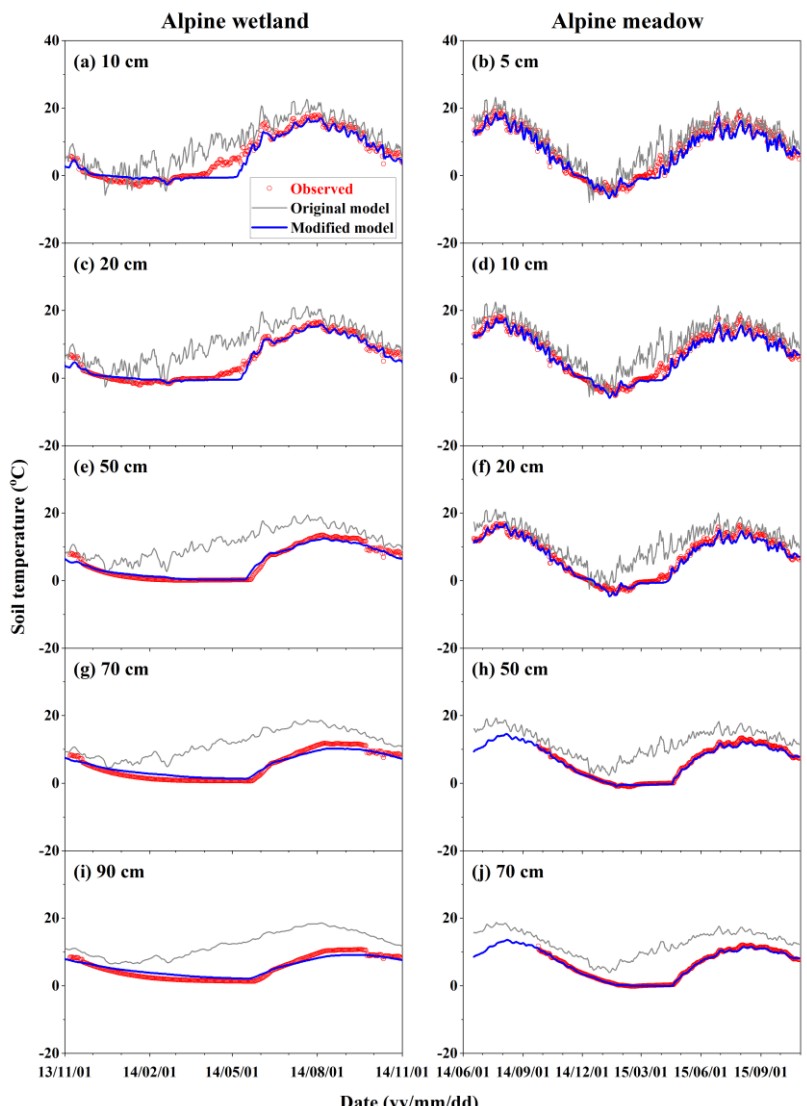

**Figure: 1 Observed and simulated daily profile soil temperature from the alpine wetlands and meadows by the original and**
**modified models. The legends in panel a apply for all panels.**

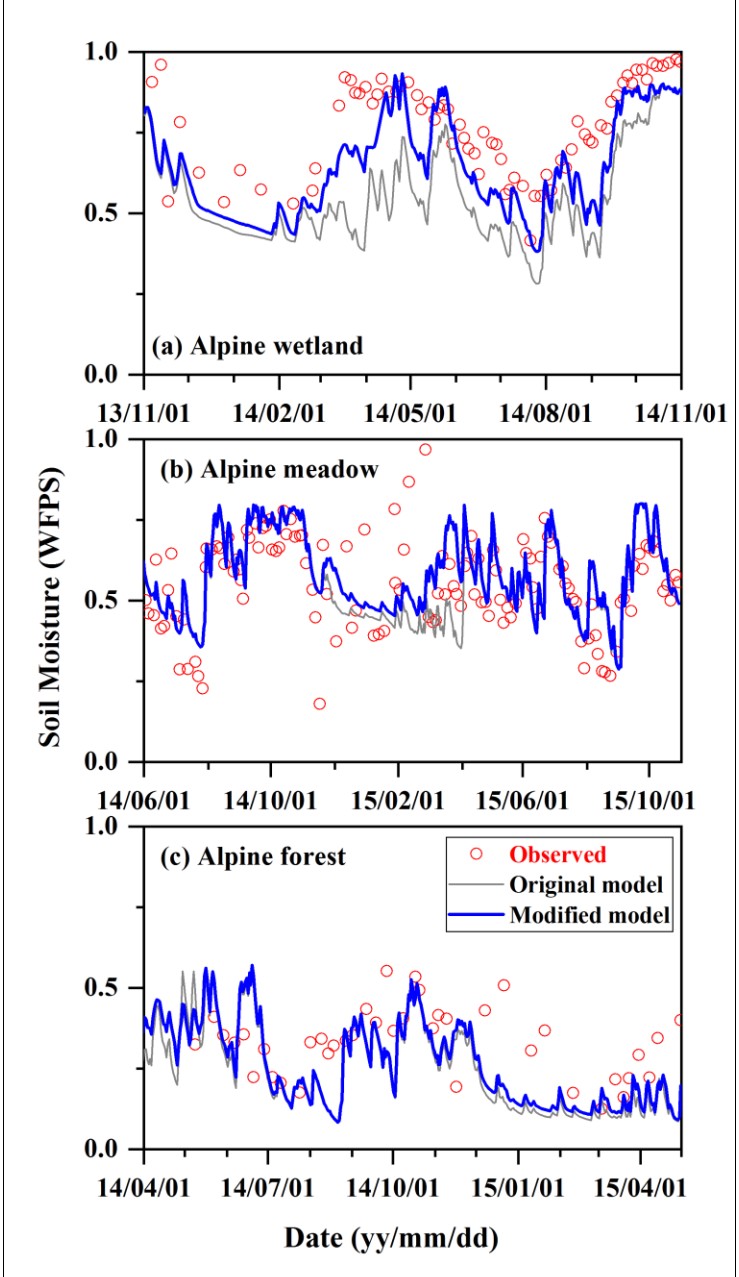

**Figure: 2 Observed and simulated daily topsoil (0–6 cm) moisture in the water-filled pore space (WFPS) from the alpine wetlands,**
**meadows and forests by the original and modified models. The legends in panel a apply for all panels.**

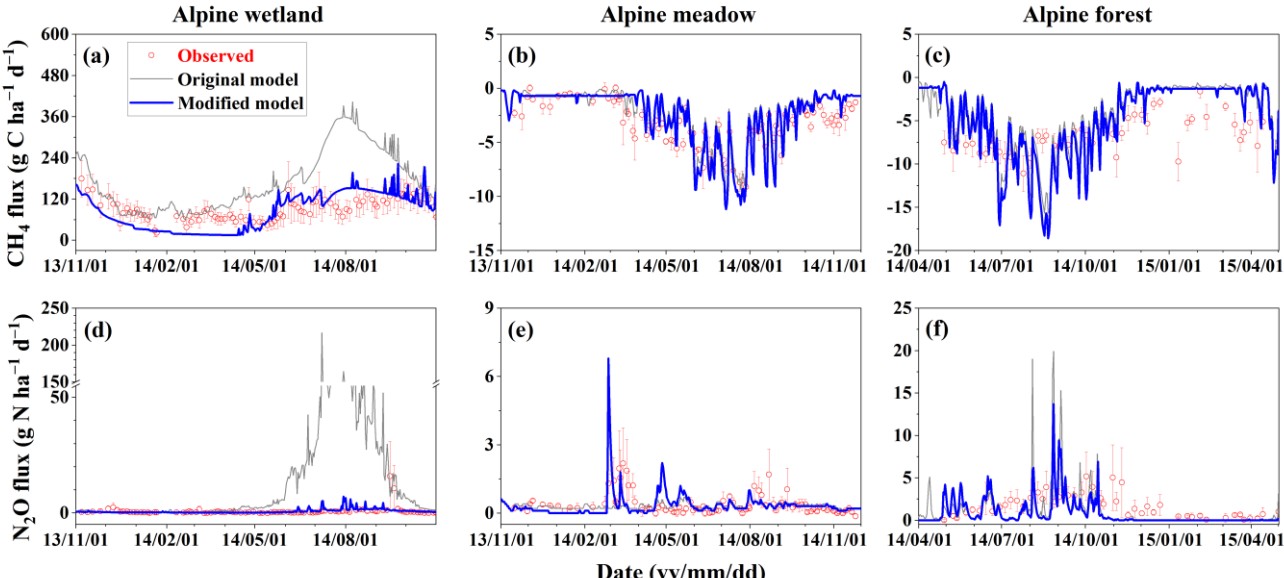

**Figure: 3 Observed and simulated daily methane (CH₄) and nitrous oxide (N₂O) fluxes from the alpine wetlands, meadows and**
**forests by the original and modified models. The vertical bar for each observation indicates the standard error of six spatial**
**replicates. The legends in panel a apply for all panels.**

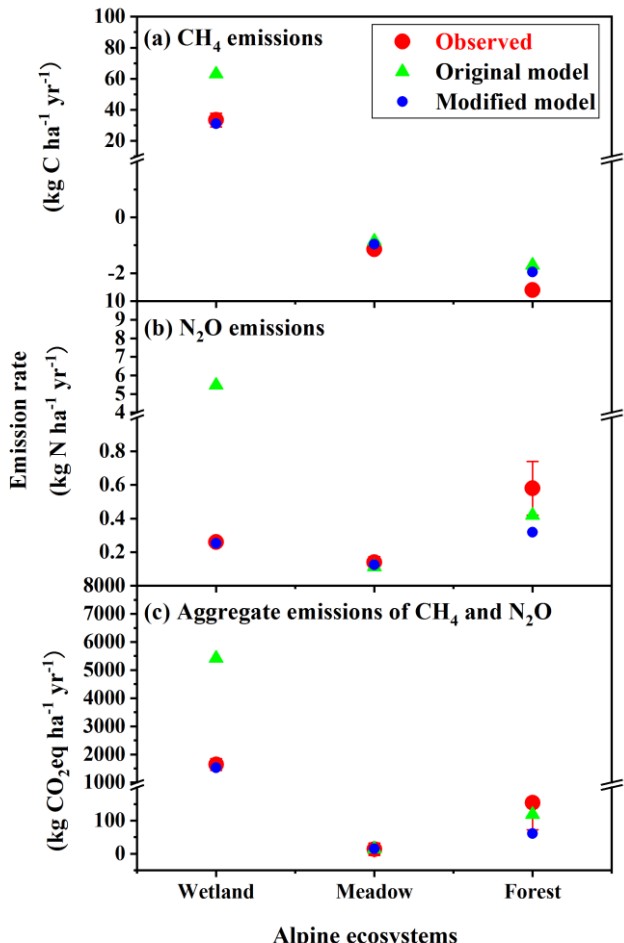


Figure: 4 Observed and simulated annual emissions of methane ($CH_4$), nitrous oxide ($N_2O$) and aggregate emissions of both from
the alpine wetlands (Wetland), meadows (Meadow) and forests (Forest). The legends in panel a apply for all panels.