# Peer review of "An improved process-oriented hydro-biogeochemical model for simulating dynamic fluxes of methane and nitrous oxide in alpine ecosystems with seasonally frozen soils"

_Biogeosciences, 2020_

## Referee Comment (RC2)

**Comments about paper "An improved process-oriented hydro-biogeochemical model for simulating dynamic fluxes of methane and nitrous oxide in alpine ecosystems with seasonally frozen soils" Submitted by Zhang et al. to Biogeosciences**

**General remarks**

This paper presents measurements and modelling of evolution of soil temperature, soil moisture, methane and nitrous oxide emissions in cold altitude wetlands, alpine meadows and forest. The used CNMM-DNDC model has been adapted by the authors to simulate the freeze-thaw cycles of the area. It is a matter for Biogeosciences but the authors have to improve seriously the paper before publication.

Primarily, they have to complete their bibliographic list and more discuss their results considering other published models. CNMM-DNDC incorporates the core biogeochemical processes of DNDC into the hydrological framework of CNMM (Catchment Nutrient Management Model). These coupling attempts, like present adaptation to freeze-thaw cycles are interesting efforts. My doubts concern the use without discussion of the DNDC corpus published near 30 years ago, when other models have showed more recently their interest in modelling the ecological functioning of microorganisms, using a smaller number of well-defined kinetic parameters and avoiding badly defined parameters of flow fractionation. Conversely, the decomposition part of DNDC uses a lot of these parameters and is for me largely over parameterized. Consequently, its prediction must be discussed, not only by the results of the usual statistic tests, comparing measurements and predictions (It is well known that any complex signal can be adjusted by any model using a great number of parameter e.g. in Fourrier transforms), but also in terms of microbial functioning.

Also the article needs to be improved by more explanations about the CNMM part. Some surprising choices must be better justified in the text, like why to predict temperature in all the profile for two systems when moisture is described in the 3 systems but only in the surface layer. The particular remarks below must be taken into account.

**Particular remarks**

L37 Is this 1st sentence a little banal?

L 60-62 the listed biogeochemical models is not exhaustive and limited to relatively old models sometimes subjected to critics (over parameterization, functional role of microorganisms…), e.g. the model MOMOS is ignored, please complete the list in such a way to better situate your work in the literature data.

2.1 paragraphs: the original paper on DNDC includes a sub model for denitrification but not for methane emission; please clarify this part. The only denitrification submodel includes 17

parameters which must be combined to the numerous parameters regulating various splitting's in the DNDC core! By two points you can adjust a right line, but also a parabola, then more and more sophisticated functions by increasing the number of parameters, and the fittings will be already OK when evaluated by the statistic laws. All the models will predict the two points. You must discuss about the number of parameters in the paper (Ockham razor)

L111-112 OK for hydro-geochemical but there is a doubt for biogeochemical, please discuss more deeply your choices considering all the literature data. The same for biogeochemistry in line 112; in contrast, paragraph 2 of 2.1.1 explain a particular interest of CNMM-DNDC

Eq5: It looks like a sum, not a weighted average? Should $j$ be defined in L148?

Eq.6: product or geometric mean?

L148-151: is there references data for the values of C and k for each constituent $j$?

L168: reference for this opaque chamber method? Is not a risk of perturbation by elimination of solar radiation?

L189-190: could the layer description be clarified?

Table1: Could the legend remember the meaning of IA, NSI and R2? I suppose P is the probability of rejection, but for what test? Are the P columns necessary since P is always <0.001?

Fig.1: legend in 1a is not very clear, perhaps write "observed" in red color; the same for other Figs

L210: ZIR does not appear in table 1

L216: The range values does not correspond to that of Table 1, please clarify

L218: where is Fig S2? (the same for all S figures, I suppose supplementary not visible for me)

Improve coherencies in Figs or more explain your choices: Fig 1 shows soil temperature for two systems at all soil layers; in contrast, Fig.2 shows moisture in the 3 systems but only in the 0-6 cm layer

Fig3: format date of x axis not very clear

L223: is the term "Water movement" exaggerated when you speak only of the surface water?

Fig.6:y axis legend not clear

L293-294: should you present succinctly this concept of the CH4 balloon in material and methods?

L351: define TP

| Principal criteria |
| --- |

**Scientific significance:**
Does the manuscript represent a substantial contribution to scientific progress within the scope of Biogeosciences (subst
new concepts, ideas, methods, or data)?

yes

**Scientific quality:**
Are the scientific approach and applied methods valid? yes

Are the results discussed in an appropriate and balanced way (consideration of related work,

including appropriate references)? No

**Presentation quality:**
Are the scientific results and conclusions presented in a clear, concise, and well-structured way (number and quality of
figures/tables, appropriate use of English language)? No

**Access review (quick report), peer review, and interactive public discussion (BGD)**

Manuscripts submitted to BG at first undergo a rapid access review (initial manuscript evaluation), which is not meant to be a full scientific review but to identify and sort out manuscripts with obvious major deficiencies in view of the above principal evaluation criteria.

Manuscripts rated 4 (poor) in any of the principal criteria are normally rejected without further review and discussion. Manuscripts rated 1–3 (excellent–fair) in all criteria are normally published on the Biogeosciences Discussions (BGD) website, the discussion forum of BG, where they are subject to full peer review and interactive public discussion.

In the full review and interactive discussion, the referees and other interested members of the scientific community are asked to take into account all of the following aspects:

1. Does the paper address relevant scientific questions within the scope of BG? Yes

2. Does the paper present novel concepts, ideas, tools, or data? New data

3. Are substantial conclusions reached? No

4. Are the scientific methods and assumptions valid and clearly outlined? Incomplete (see my remarks)

5. Are the results sufficient to support the interpretations and conclusions? yes

6. Is the description of experiments and calculations sufficiently complete and precise to allow their reproduction by fellow scientists (traceability of results)? No

7. Do the authors give proper credit to related work and clearly indicate their own new/original contribution? Yes

8. Does the title clearly reflect the contents of the paper? yes

9. Does the abstract provide a concise and complete summary? no

10. Is the overall presentation well structured and clear? No, see my remarks above

11. Is the language fluent and precise? yes

12. Are mathematical formulae, symbols, abbreviations, and units correctly defined and used? No, see my remarks

13. Should any parts of the paper (text, formulae, figures, tables) be clarified, reduced, combined, or eliminated? yes

14. Are the number and quality of references appropriate? No

15. Is the amount and quality of supplementary material appropriate?

**Peer-review completion (BG)**

At the end of the interactive public discussion, the authors may make their final response and submit a revised manuscript. Based on the referee comments, other relevant comments, and the authors' response in the public discussion, the revised manuscript is re-evaluated and rated by the associate editor. If rated **excellent** or **good** in all of the principal criteria and specific aspects listed above, it will normally be accepted for publication in BG. Additional advice from the referees in the evaluation and rating of the revised manuscript will be requested by the associate editor if the public discussion in BGD is not sufficiently conclusive.

Biogeoscie

---

## Author Response (AR1)

The submitted article by Wei Zhang et al. present the new module to better describe the thermal dynamics within the hydro-biogeochimical model CNMM-DNDC. The authors set up a catchment scale modelling approach and test it on observed soil temperature, water filled pore space, CH4 and N2O emissions from three alpine ecosystems of the Tibetan Plateau. The authors conclude that their proposed module improves the reliability of all investigated measured criteria. The manuscript is well written and the results prove the benefit of the modified modelling approach.

General Comments

Starting very general, I doubt that a default model setup can be used to determine, whether a model is capable to reproduce a measured criteria or not. As such, the presented simulations of the "original" as well as "modified" model are, to me, just random, given the vast amount of internal model parameters. In my opinion both models would need a prior calibration to the local data set first, before judging whether the model is capable to reproduce the observed criteria or not. Maybe there is a parameter combination in the original mode, which performs much better than the modified version. As such, I would reject this manuscript given this general point. However, I leave this point to the Editor, as I know that it is still common to investigated not calibrated models in biogeochemical sciences. In the hydrological community, it is not. This paper covers both disciplines.

**Revised**.

We fully agreed with the reviewer. As a result, the prior calibration issue was raised in the section of discussion.

"Compared with the empirical model, one key advantage of the process-oriented models is that the models are independent of the local parameterization (Zhang *et al.*, 2015). In this study, default internal parameter combinations of biogeochemical processes were used for the original and modified models, which have been applied in the catchment simulation in the subtropical region (Zhang *et al.*, 2018b), due to the limit field observations (only one year) for both calibration and validation. The biogeochemical processes were predicted by the first-order and Michaelis-Menten kinetics in the CNMM-DNDC based on some defined parameters of flow fractionation. For instance, there are 17 parameters related with $N_2O$ emission in the module of denitrification (Table S6), which would inevitably increase the uncertainty of simulation. Houska *et al.* (2017) found that hydro-biogeochemical models can be right for the wrong reasons, such as matching greenhouse gas emissions while failing to simulate soil moisture, which emphasized the importance for simultaneous validations of multi-variables. Thus, simultaneous validations of $CH_4$ and $N_2O$ fluxes, as well as soil environment variables, were necessary for comprehensive evaluation of the model performance. In addition, the microbial ecology was recently recommended to be integrated into the biogeochemical model using a smaller number of well-defined kinetic parameters, such as MOMOS (Pansu *et al.*, 2010; Treseder *et al.*, 2011). Therefore, the direct control of microbial on biogeochemical processes, such as the stoichiometry of decomposer, is required to be included in the CNMM-DNDC in near

future." (**See lines 397-410 in the revised manuscript**).

Another general point, which makes the judgement of this manuscript difficult for me is, that neither the model, nor the observation data, nor the model setups are accessible to me. Given the open access policy of BG, I was quite surprised to see that. Under these circumstances and given the not very detailed Materials and Methods section, which cites a lot, but gives very view details, I had to guess a lot. I made these guesses within this review in favor to the authors of this manuscript, assuming scientific correctness, without being able as a reviewer to check. E.g., if the given equations proper implemented in modelling code or if the model was correct set up for the characteristics of the study catchment. I feel a lot of improvement necessary within this manuscript to give the readers the possibility to understand and reproduce the results of this study. This point is not easy to cover given the current stage of the manuscript. I leave also this second general point to the Editor, to decide, whether it is reason to reject the manuscript.

**Revised**.

We fully agreed with the reviewer. The availability of observed data has been stated in the Data availability section. The detailed information of model setups has been added in Tables S1−4 and the section of Materials and methods.

The detailed information about the hydrological model of CNMM has been added.
"... into the hydrological framework of the CNMM, which is fully distributed." (**See lines 120-121 in the revised manuscript**). "The soil moisture was calculated based on the mass balance of precipitation, irrigation, evapotranspiration, vertical flow, lateral flow and water from a rising water table. The total water that can be infiltrated during each time step was determined by a defined maximum infiltration rate. Darcy's law was applied for predicting the vertical water flow in the soil profile. A cell-by-cell approach using a kinematic approximation was applied to route the saturated overland and subsurface flow based on DEM. The stream flow was estimated using a cascade of linear channel reservoirs (Wigmosta *et al.*, 1994)." (**See lines 130-135 in the revised manuscript**).

Detailed statements about the differences of the original and modified model in the soil thermal module, as well as the equation used in the original model (Table S1), have been added.
"In the CNMM-DNDC, the soil temperature was predicted by solving the one-dimensional heat conduction equation with the implicit method of Crank-Nicholson. However, despite the simple parameterization used for the calculation of soil heat capacity and thermal conductivity, the variations of soil temperature induced by the freeze-thaw cycles were also not considered (Table S1 of the online supplementary materials), which inevitably hindered its application in the seasonally frozen regions. In this study, the CNMM-DNDC was modified by replacing the above soil thermal module by a physical based module of Northern

Ecosystem Soil Temperature (Zhang *et al.*, 2003; Deng *et al.*, 2014), which can explicitly describe the energy exchange within the soil, the active layer dynamics and the soil thermal regime in the presence of freeze-thaw cycles. ... Therefore, the CNMM-DNDC with and without the above modifications is hereafter referred to as the original and modified model, respectively." (**See lines 152-162 in the revised manuscript**). "Compared to the original thermal module, the internal heat exchange due to freezing or thawing (*S*) was included with improved algorithm for thermal conductivity (*k*). In addition, the one-dimensional heat conduction equation (Eq. 1) was solved by converting it to an explicit form in the modified model (Eqs. 2–4), while was solved with the implicit method in the original models (Table S1)." (**See lines 181-184 in the revised manuscript**).

The reason for the catchment simulation was added.
"For the target catchment, the soil water dynamics of the alpine ecosystems were determined by the precipitation, evapotranspiration, infiltration, penetration and lateral flow. Using the database, a catchment simulation of hydro-biogeochemical processes was performed with spatial and temporal resolutions of $30 \times 30$ m$^2$ and 3 hours, respectively, by the modified CNMM-DNDC from 2012 to 2015, which could reflect the influences of hydrological processes on soil water dynamics. Thus, the soil water dynamics of the seasonally inundated wetlands were determined by the hydrological processes without any artificial disturbances in the catchment simulation." (**See lines 225-230 in the revised manuscript**).

The detailed the calculation steps of soil heat capacity ($C_l$) and thermal conductivity ($k_l$), as well as the related contents about the scenario simulation of annually inundated (A-wetland) wetland have been added. See the responses below, please.

My further comments can be seen as major revision, which will hopefully help the authors to improve their manuscript, either for future publication in this journal or after potential rejection in another journal.

Specific comments

Table 1: Please specify "n" in Table caption. Please add a column for data resolution. Regarding CH4 an N2O fluxes, Tables say its daily, while Text (Line 167) says its weekly. Further, statistical indices (IA, NSI, Slope, R $^2$ and P) are not in line with chapter 2.4 (IA, NSE, R $^2$, ZIR and MRB). Also, give please give the full name of the statistical indices in the Table caption. Use NSE (Nash-Sutcliffe efficiency) instead of NSI throughout the manuscript. How is it possible that there are no values for R $^2$ and P, if you could calculate IA, NSI and Slope? Please give the equations for ZIR and MRB in chapter 2.4, as there are not so commonly used.
**Revised**.
Table 1 has been revised as the reviewer suggested by adding detailed notes. The

equation applied for the zero-intercept univariate linear regression (ZIR) of the observations against the simulations has been added in Table S5. The no values for $R^2$ were due to the sum of regression square are larger than the sum of the total square for the zero-intercept univariate linear regression, which has been added in the note of Table 1. (**See Table 1 in the revised manuscript**).

Figure 1: In general, a good simulation of the observed soil temperature is nothing special in the state of the art environmental models. The "original" model just seems to be wrong, so it is not a big challenge to improve that. However, still a necessary task, which shows the importance of this work. What I am missing in the methods chapter, is a clear description of the differences between the "modified" and the "original" model. Maybe a figure would help, showing the setup of both models and highlighting the differences.

**Revised**.

Detailed statements about the differences of the original and modified model in the soil thermal module, as well as the equation used in the original model (Table S1), have been added to make it clear.

"In the CNMM-DNDC, the soil temperature was predicted by solving the one-dimensional heat conduction equation with the implicit method of Crank-Nicholson. However, despite the simple parameterization used for the calculation of soil heat capacity and thermal conductivity, the variations of soil temperature induced by the freeze-thaw cycles were also not considered (Table S1 of the online supplementary materials), which inevitably hindered its application in the seasonally frozen regions. In this study, the CNMM-DNDC was modified by replacing the above soil thermal module by a physical based module of Northern Ecosystem Soil Temperature (Zhang *et al.*, 2003; Deng *et al.*, 2014), which can explicitly describe the energy exchange within the soil, the active layer dynamics and the soil thermal regime in the presence of freeze-thaw cycles. ... Therefore, the CNMM-DNDC with and without the above modifications is hereafter referred to as the original and modified model, respectively." (**See lines 152-162 in the revised manuscript**). "Compared to the original thermal module, the internal heat exchange due to freezing or thawing ($S$) was included with improved algorithm for thermal conductivity ($k$). In addition, the one-dimensional heat conduction equation (Eq. 1) was solved by converting it to an explicit form in the modified model (Eqs. 2–4), while was solved with the implicit method in the original models (Table S1)." (**See lines 181-184 in the revised manuscript**).

Line 221-224: This statement is way to bold. Firstly, you tested the model in one region only. Secondly, the models best achieved NSI value is 0.32. That's very far from a "reliable" prediction (please add some citations for comparison in the Discussion chapter, there are tons, e.g. Ford et al., 2014). Thirdly, you tested the model on top soil WFPS only. How can you generalize from there to "reliable water movement" in

general? Please delete this passage and be more accurate throughout the manuscript.
**Revised**.
More discussions have been added in relation with the simulated soil temperature and moisture with references.

"In order to quantify the impacts of climate change on the cycling of carbon and water on the regional and global scales, several large scale ecosystem models or macroscale hydrological models, such as Terrestrial Ecosystem Model, Lund-Potsdam-Jena dynamic global vegetation model and Variable Infiltration Capacity model, have been enhanced to simulate the soil thermal dynamics at northern high latitude (Wania *et al.*, 2009; Zhuang *et al.*, 2001; Cuo *et al.*, 2015; Jiang *et al.*, 2020). In addition, the soil thermal modules were also improved in some biogeochemical models, such as DNDC and Mobile-DNDC, to evaluate the influences of climate warming on the biogeochemical processes in high latitude regions (Zhang *et al.*, 2003; de Bruijn *et al.*, 2009; Wolf *et al.*, 2011; Zhang *et al.*, 2012; Deng *et al.*, 2014). Compared with the simulated soil profile temperatures by above models at different scales, the simulations in this study by the modified CNMM-DNDC were equally well, especially for deeper soil layers (e.g., Wania *et al.*, 2009). For the validated topsoil moisture in this study, the modified model generally captured the variation trends, which were comparable with the performances of other models (e.g., de Bruijn *et al.*, 2009; Wolf *et al.*, 2011; Cuo *et al.*, 2015). However, compared with the studies focused on simulating soil moisture (e.g., Ford *et al.*, 2014), further improvements are still required to improve the model performance in simulating the soil moisture." (**See lines 316-328 in the revised manuscript**).

The statements have been revised throughout the manuscript (**See lines 261-264, 277-278, 292-293, 328-330, 410-413 and 425-427 in the revised manuscript**).

Line 235-237: I do not understand how the results show that the model "simulated the CH4 fluxes [..] at the catchment scale". The set up might be fully distributed (which is a guess by me and not very clear stated in the Material and Methods section). However, you test the model on local scale measurements. So, you can only state that the model is able to reproduce the local measurements with the given statistical accuracy. Please be consistent with this comment throughout the manuscript. Further, I do not understand why a fully-distributed model set-up is needed herein to test the local measurements as you do not have any spatial measurements. If it is needed, it needs some justification within the manuscript and some results on the spatial scale, e.g. a map of the N2O and CH4 emissions.
**Revised**.
We fully agree with the reviewer. The detailed information about the hydrological model of CNMM has been added.

"... into the hydrological framework of the CNMM, which is fully distributed." (**See lines 120-121 in the revised manuscript**). "The soil moisture was calculated based on the mass balance of precipitation, irrigation, evapotranspiration, vertical flow,

lateral flow and water from a rising water table. The total water that can be infiltrated during each time step was determined by a defined maximum infiltration rate. Darcy's law was applied for predicting the vertical water flow in the soil profile. A cell-by-cell approach using a kinematic approximation was applied to route the saturated overland and subsurface flow based on DEM. The stream flow was estimated using a cascade of linear channel reservoirs (Wigmosta *et al.*, 1994)." (**See lines 130-135 in the revised manuscript**).

The reason for the catchment simulation, as well as the simulation results, was added. "For the target catchment, the soil water dynamics of the alpine ecosystems were determined by the precipitation, evapotranspiration, infiltration, penetration and lateral flow. Using the database, a catchment simulation of hydro-biogeochemical processes was performed with spatial and temporal resolutions of $30 \times 30$ m$^2$ and 3 hours, respectively, by the modified CNMM-DNDC from 2012 to 2015, which could reflect the influences of hydrological processes on soil water dynamics. Thus, the soil water dynamics of the seasonally inundated wetlands were determined by the hydrological processes without any artificial disturbances in the catchment simulation." (**See lines 225-230 in the revised manuscript**). "For the catchment simulation, the simulated annual CH$_4$ emissions ranged from -2.35 to 73.0 kg C ha$^{-1}$ yr$^{-1}$ from November 2013 to November 2014 (Fig. S4a)." (**See lines 275-277 in the revised manuscript**). "For the catchment simulation, the simulated annual N$_2$O emissions ranged from 0.01 to 0.74 kg N ha$^{-1}$ yr$^{-1}$ from November 2013 to November 2014 (Fig. S4b)." (**See lines 291-292 in the revised manuscript**).

The statements have been revised throughout the manuscript (**See lines 261-264, 277-278, 292-293, 328-330, 410-413 and 425-427 in the revised manuscript**).

Figure 3: From the differences in the WFPS and soil temperature, I do not see any reason why the original model would produce such a high N2O peak and the modified model (assuming that the described soil temperature routine was the only thing changed, which is not 100% clear in chapter 2.1.2) does not explain such a vast difference in either denitrification and nitrification processes. This needs some more detailed discussion in chapter 4.3, also by showing the different model internal processes of nitrification and denitrification of the two model set ups explaining the difference.
**Revised**.
Detailed statements about the differences of the original and modified model in the soil thermal module, as well as the equation used in the original model (Table S1), have been added.
"In the CNMM-DNDC, the soil temperature was predicted by solving the one-dimensional heat conduction equation with the implicit method of Crank-Nicholson. However, despite the simple parameterization used for the calculation of soil heat capacity and thermal conductivity, the variations of soil temperature induced by the freeze-thaw cycles were also not considered (Table S1 of

the online supplementary materials), which inevitably hindered its application in the seasonally frozen regions. In this study, the CNMM-DNDC was modified by replacing the above soil thermal module by a physical based module of Northern Ecosystem Soil Temperature (Zhang *et al.*, 2003; Deng *et al.*, 2014), which can explicitly describe the energy exchange within the soil, the active layer dynamics and the soil thermal regime in the presence of freeze-thaw cycles. ... Therefore, the CNMM-DNDC with and without the above modifications is hereafter referred to as the original and modified model, respectively." (**See lines 152-162 in the revised manuscript**). "Compared to the original thermal module, the internal heat exchange due to freezing or thawing ($S$) was included with improved algorithm for thermal conductivity ($k$). In addition, the one-dimensional heat conduction equation (Eq. 1) was solved by converting it to an explicit form in the modified model (Eqs. 2–4), while was solved with the implicit method in the original models (Table S1)." (**See lines 181-184 in the revised manuscript**).

The detailed information of model setup parameters has been added in Table S3. (**See Table S3**).

The discussion about the intensive $N_2O$ emissions has been added. "The intensive $N_2O$ emissions simulated by the original model resulted from the overestimated soil temperature for the alpine wetlands. Firstly, as the presence of ice could impede the water movement, the water lateral flows were promoted by the original model due to the neglecting of freeze-thaw cycles. These further resulted in the lower simulated soil moisture as compared with the modified model (Fig. S5), which provided favorable oxygen conditions for $N_2O$ production. Meanwhile, the simulated high soil moisture by the modified model provided feasible anaerobic conditions for thoroughly denitrification. Secondly, higher simulated soil temperature by the original model also facilitated the mineralization, which provided more available mineral nitrogen." (**See lines 367-374 in the revised manuscript**).

Line 344-347: Again a very bold statement, speaking of "hydrology" when testing a model with WFPS measured at the upper 6 cm of the soil (what about evaporation, overland flow, infiltration, groundwater recharge,…); speaking of "nitrogen cycling" when testing against N2O emission, which is only 1% of the total N emissions (what about N2, NH3, NO, the nitrogen stored in the soil,…), speaking of "carbon cycling" while not looking at CO2 emissions or the changing carbon storage. Please rephrase and stick to the investigated processes throughout the manuscript.
**Revised**.
The statements have been revised throughout the manuscript (**See lines 261-264, 277-278, 292-293, 328-330, 410-413 and 425-427 in the revised manuscript**).

Section 4.4: Interesting section, however, atm out of the scope of the manuscript (which implies so far the testing of a changed module in a model). If this section is supposed to

be included in the manuscript, please extend, title, abstract, methods section and add a figure are table to visualize this discussion. I would recommend to delete.

**Revised**.

The related contents have been deleted as the reviewer suggested.

Please add a comparison of the model results with other studies investigating wetlands, meadows, forest. Also a comparison to different models, which implemented and tested thermal dynamics in their models. And what about other studies investigating freeze thaw cyclings with hydro-biogeochemical models, e.g. deBruijn et al (2009). I am surprised not to see many of the relevant literature within the discussion, please add.

**Revised**.

We fully agreed with the reviewer. More discussions have been added in Discussion section with references.

"For the studies focused on simulating $CH_4$ emissions from wetlands by the large-scale ecosystem models, the model validation with field observation is difficult due to coarse spatial resolution (e.g., Zhuang *et al.*, 2004). For the biogeochemical model, such as DNDC, the dynamics of $CH_4$ emissions from wetland and peatland in the northern permafrost regions were well simulated (Zhang *et al.*, 2012; Deng *et al.*, 2014), which showed consistent seasonal variations and magnitudes as those in this study." (**See lines 345-349 in the revised manuscript**). "Field observations showed that the soil freeze-thaw cycles occurred in seasonally frozen regions not only increased the availability of nitrogen and carbon substrates by disrupting of soil aggregates but also affected the structure, population and activity of the microbes, and thus influencing the emissions of $N_2O$ (e.g., Song *et al.*, 2019). de Bruijn *et al.* (2009) have explored the combined mechanisms for simulating freeze–thaw related $N_2O$ emissions, which were the promoted anaerobiosis and denitrification due to reduced gas diffusion derived from soil frost and snow cover, and the stimulated microbial growth due to easy decomposable organic carbon and nitrogen derived from the dead microbes during freeze-thaw cycles. Wolf *et al.* (2011) introduced an impedance factor to parameterize the reduced water flow between layers in presence of ice, which could captured the freeze–thaw related $N_2O$ emissions for ungrazed steppe. In the CNMM-DNDC, threshold values of soil temperature were set to trigger the death of microbes during the freezing period and stimulate the production of NO, $N_2O$ and $N_2$ using substrates derived from the dead microbes during the thawing period, which was similar to one of the mechanisms explored by de Bruijn *et al.* (2009). However, compared with the simulated freeze–thaw related $N_2O$ emissions by other studies, the simulated dynamics of peak emissions due to freeze-thaw cycles in this study were inconsistent with those from the field observations. Thus, improvements are required to incorporate some other effective mechanisms to better capture the dynamic characteristics." (**See lines 379-392 in the revised manuscript**).

Technical corrections

Line 22: Change "as" to "is"

**Revised**.

The sentence has been revised as the reviewer suggested (**See lines 21-23 in the revised manuscript**).

Line 34: I don't understand how the model can be used to "evaluate the sustainability". Please rephrase.

**Revised**.

The sentence has been rewritten to make it clear.

"As the original CNMM-DNDC was previously validated in some unfrozen regions, the modified CNMM-DNDC could be potentially applied to estimate the emissions of $CH_4$ and $N_2O$ from various ecosystems under different climate zones at the site or catchment scale." (**See lines 32-34 in the revised manuscript**).

Line 44: Expression "during long periods" please be more accurate and give some numbers.

**Revised**.

The sentence has been revised (**See lines 44-45 in the revised manuscript**).

Line 125: Needs a reference to the concept.

**Revised**.

The reference has been added (**See lines 141-142 in the revised manuscript**).

Line 134-136: Please simplify structure of the sentence

**Revised**.

The sentences have been rewritten (**See lines 152-158 in the revised manuscript**).

Line 134-140: It remains unclear to me, whether these changes are already done in the cited publications, or if that's the new part.

**Revised**.

This part has been rewritten to make it clear (**See lines 152-162 in the revised manuscript**).

Line 149: What are these numbers behind organic matter, mineral, water, ice and air? I have to assume they are dynamic for each time step.

**Revised**.

These sentences have been revised.

"The dynamic soil heat capacity ($C_l$, J m$^{-3}$ ℃$^{-1}$) is the weighted average of the heat capacity for five constituents, including organic matter ($C_{l, OM}$), minerals ($C_{l,Min}$), water ($C_{l, Water}$), ice ($C_{l, Ice}$) and air ($C_{l, Air}$) (Eq. 5). The values of heat capacity for organic matter, minerals, water, ice and air were $2.5 \times 10^6$, $2.0 \times 10^6$, $4.2 \times 10^6$, $2.1 \times 10^6$ and $1.2 \times 10^3$ J m$^{-3}$ ℃$^{-1}$, respectively (Huang, 2000)." (**See lines 170-173 in the revised manuscript**).

Equations 1-6: Why not add an example with a given organic matter, minerals, water ice and air values.
**Revised**.
The detailed the calculation steps of soil heat capacity ($C_l$) and thermal conductivity ($k_l$) have been added as the reviewer suggested. (**See Eq. 5–13 in the revised manuscript**).

Line 153: Why in 35 m? Is that a constant? Or the depth to which the model can be applied to?
**Revised**.
The sentences have been revised to make it clear.
"The simulated soil depth (including bedrock) is user-defined." (**See line 129 in the revised manuscript**). "The simulated soil depth was defined as 35 m due to the lower boundary conditions of the thermal dynamics, which was set as the geothermal heat flux at a soil depth of 35 m. The simulated soil profile (0−35 m depth) was divided into 23 layers, including the soil (0−1.5m) and bedrock (1.5−35m)." (**See lines 219-222 in the revised manuscript**).

Line 175: Details about the instruments used for the soil temperature and WFPS measurements are relevant within this manuscript, please add.
**Revised**.
The detailed instruments used for field measurements have been added in the Table S2 of the online supplementary materials. (**See Table S2 in the revised manuscript**).

Line 188: Which pedo-transfer function?
**Revised**.
The applied pedo-transfer functions were detailed in the Table S4 of the online supplementary materials. (**See Table S4 in the revised manuscript**).

Line 192: Why was the model run in 3 hours resolution if the metrological data input is hourly available?
**Revised**.
The time resolution is determined by both the model and meteorological data, which

were at hour scale during field observation, but daily scale at the other period.

 "The temporal and spatial resolutions are also user-defined according to the driving data of climate (generally in 3 hours) and digital elevation model (DEM)." (**See lines 129-130 in the revised manuscript**).  "a climate dataset of 3-hour weather data (air temperature, precipitation, wind speed, solar radiation, longwave radiation, and humidity), which were obtained from the meteorological station in the target catchment for the years with field observations (2013.11–2015.10) and were adapted from the daily data at the Zoige Meteorological Station (provided by the National Meteorological Information Center: http://data.cma.cn/; last access: 10th June, 2020) for other years;" (**See lines 211-215 in the revised manuscript**).

Figure 2: Please use same y-axis range throught subplots. It is odd to have WFPS up to 1.2. Please use range from 0-1.
**Revised**.
The figure has been revised as the reviewer suggested. (**See Fig. 2 in the revised manuscript**).

Figure 1-5: It would be way easier to interpret if there would be one figure for each land use with all the fluxes. Maybe even only one figure including all fluxes.
**Revised**.
The figures have been adjusted to make the fluxes of $CH_4$ and $N_2O$ from all alpine ecosystems in one figure. (**See Fig. 3 in the revised manuscript**).

Line250-252: Again, a too broad generalization from one model run and one study area. Please stick with the expression to the investigated processes, e.g. "These results indicate that the modified CNMM-DNDC has the potential to estimate N2O emissions in a seasonally frozen region."
**Revised**.
The sentence has been revised as the reviewer suggested (**See lines 292-293 in the revised manuscript**).

Figure 6: Why is there now a second wetland (A-wetland) which hasn't been shown before? Please stay consistent. Again it would be easier to group the study areas, as the results here are hard to read. Maybe a table would help to.
**Revised**.
The related contents have been deleted as the reviewer suggested.

Line 262-272: Please move to the discussion chapter. Also, I see what the authors want to express here, however, investigating the climate impact from different landuse is not

expressed as scope of the manuscript.

**Revised**.

The related contents have been deleted as the reviewer suggested.

Line 275-278: I would have liked to see some more insight into the model internal processes and differences here. E.g. a picture showing the different soil layers in the models and compare the WFPS, maybe similar to Figure 12 in Haas et al (2013) or Figure 7 in Klatt et al (2017).

**Revised**.

The Fig. S5 has been added to show the simulated soil profile moisture of the alpine meadow by the original and modified models, which resulted in the different simulation of $N_2O$ emissions.

"The intensive $N_2O$ emissions simulated by the original model resulted from the overestimated soil temperature for the alpine wetlands. Firstly, as the presence of ice could impede the water movement, the water lateral flows were promoted by the original model due to the neglecting of freeze-thaw cycles. These further resulted in the lower simulated soil moisture as compared with the modified model (Fig. S5), which provided favorable oxygen conditions for $N_2O$ production. Meanwhile, the simulated high soil moisture by the modified model provided feasible anaerobic conditions for thoroughly denitrification. Secondly, higher simulated soil temperature by the original model also facilitated the mineralization, which provided more available mineral nitrogen." (**See lines 367-374 in the revised manuscript**).

Line 304: Needs a reference where global warming effect on CH4 emissions where investigated.

**Revised**.

The reference of a review has been added (**See lines 343-345 in the revised manuscript**).

Line 305-306: To be able to understand this sentence, this manuscript needs a table showing the relevant model setup up parameters (meteorology, soil, management, vegetation). Maybe something similar as Table 1 in Houska et al (2017).

**Revised**.

The detailed information of model setup parameters has been added in Table S3. (**See Table S3 in the revised manuscript**).

Line 311: How was the influence of the clay fraction in the CH4 uptake investigated? Interesting point, but this statement comes out of the blue, as it was not part of the Methods and the Results sections.

**Revised**.

The reason of high CH$_4$ uptake in the forest has been rewritten to make it accurate.

"Both observations and simulations showed that the CH$_4$ uptake in alpine forests was higher than that in alpine meadows, which was mainly attributed to the high SOC content of the alpine forests in the simulation. Methane uptake by upland soils is a biological process governed by the availability of CH$_4$ and oxygen as well as the activity and quantity of methanotrophic bacteria in soils (e.g., Liu *et al.*, 2007; Zhang *et al.*, 2014). In the model, the simulated CH$_4$ uptake was positively related to the SOC content, which is closely related to the population size of methanotrophic bacteria. Thus, the SOC content primarily contributed to the differences in CH$_4$ uptake from alpine meadows and forests, as the values for forests were more than twice of those for meadows (Table S3)." (**See lines 349-355 in the revised manuscript**).

Line 328: How did the authors achieve and control an inundation in the model?

**Revised**.

The detailed information about inundation has been added.

"For the target catchment, the soil water dynamics of the alpine ecosystems were determined by the precipitation, evapotranspiration, infiltration, penetration and lateral flow. Using the database, a catchment simulation of hydro-biogeochemical processes was performed with spatial and temporal resolutions of 30×30 m$^2$ and 3 hours, respectively, by the modified CNMM-DNDC from 2012 to 2015, which could reflect the influences of hydrological processes on soil water dynamics. Thus, the soil water dynamics of the seasonally inundated wetlands were determined by the hydrological processes without any artificial disturbances in the catchment simulation." (**See lines 225-230 in the revised manuscript**).

Line 335-336: I do not understand this sentence. Is the process of "disruption of soil aggregates" as well as the "structure, population and activity of the microbes" really incuded in CNMM-DNDC. The materials and methods section is missing a description of the relevant included process. And I assume, these processes are not included, so please rephrase.

**Revised**.

The sentence has been revised to make it clear, which aimed at explaining the mechanism of peak N$_2$O emissions during freeze-thaw cycles.

"Field observations showed that the soil freeze-thaw cycles occurred in seasonally frozen regions not only increased the availability of nitrogen and carbon substrates by disrupting of soil aggregates but also affected the structure, population and activity of the microbes, and thus influencing the emissions of N$_2$O (e.g., Song *et al.*, 2019)." (**See lines 379-382 in the revised manuscript**).

Line 344: Where is the "detection limit" of the used N2O measurement technique?

**Revised**.

The detection limit of $N_2O$ measurement has been added (**See lines 395-396 in the revised manuscript**).

Line 387: Change to " implies that a hydro-biogeochemical model"

**Revised**.

The sentence has been revised (**See lines 425-427 in the revised manuscript**).

References:

A.M.G. de Bruijn, K. Butterbach-Bahl, S. Blagodatsky, R. Grote: Model evaluation of different mechanisms driving freeze–thaw N2O emissions, Agriculture, Ecosystems & Environment, 133, 196-207, https://doi.org/10.1016/j.agee.2009.04.023, 2009.

Ford, T. W., Harris, E., and Quiring, S. M.: Estimating root zone soil moisture using near-surface observations from SMOS, Hydrol. Earth Syst. Sci., 18, 139–154, https://doi.org/10.5194/hess-18-139-2014, 2014.

Haas, E., Klatt, S., Fröhlich, A. et al. LandscapeDNDC: a process model for simulation of biosphere–atmosphere–hydrosphere exchange processes at site and regional scale, Landscape Ecol, 28, 615–636, https://doi.org/10.1007/s10980-012-9772-x, 2013.

Houska, T., Kraus, D., Kiese, R., and Breuer, L.: Constraining a complex biogeochemical model for CO2 and N2O emission simulations from various land uses by model–data fusion, Biogeosciences, 14, 3487–3508, https://doi.org/10.5194/bg-14-3487-2017, 2017.

**Revised**.

These relevant references have been added which were highlighted in the Reference section.

Comments about paper "An improved process-oriented hydro-biogeochemical model for simulating dynamic fluxes of methane and nitrous oxide in alpine ecosystems with seasonally frozen soils" Submitted by Zhang et al. to Biogeosciences

General remarks

This paper presents measurements and modelling of evolution of soil temperature, soil moisture, methane and nitrous oxide emissions in cold altitude wetlands, alpine meadows and forest. The used CNMM-DNDC model has been adapted by the authors to simulate the freeze-thaw cycles of the area. It is a matter for Biogeosciences but the authors have to improve seriously the paper before publication.

Primarily, they have to complete their bibliographic list and more discuss their results considering other published models. CNMM-DNDC incorporates the core biogeochemical processes of DNDC into the hydrological framework of CNMM (Catchment Nutrient Management Model). These coupling attempts, like present adaptation to freeze-thaw cycles are interesting efforts. My doubts concern the use without discussion of the DNDC corpus published near 30 years ago, when other models have showed more recently their interest in modelling the ecological functioning of microorganisms, using a smaller number of well-defined kinetic parameters and avoiding badly defined parameters of flow fractionation. Conversely, the decomposition part of DNDC uses a lot of these parameters and is for me largely over parameterized. Consequently, its prediction must be discussed, not only by the results of the usual statistic tests, comparing measurements and predictions (It is well known that any complex signal can be adjusted by any model using a great number of parameter e.g. in Fourrier transforms), but also in terms of microbial functioning.
**Revised**.
We fully agreed with the reviewer. The descriptions about the DNDC and further discussions of the results with updated references have been added.
"Biogeochemical models, such as DNDC, LandscapeDNDC, WNMM, MOMOS, CENTURY and DayCent, are effective tools for simulating the cycling of nitrogen and carbon and quantifying the effects of climate change and anthropogenic activities on ecosystems (e.g., Foereid *et al.*, 2007; Haas *et al.*, 2012; Li, 2007; Li *et al.*, 2007; Pansu *et al.*, 2010; Cheng *et al.*, 2014; Pansu *et al.*, 2014). In recent years, some new conceptual approaches are applied in the biogeochemical models, such as centering on the functional role of the soil microbial biomass (Pansu *et al.*, 2010; Pansu *et al.*, 2014) and detailing the lateral transport of water and nutrients (Haas *et al.*, 2012; Zhang *et al.*, 2018b)." (**See lines 60-65 in the revised manuscript**). "For the new generation of biogeochemical models, the microbial ecology was integrated into the biogeochemical models, which represents direct microbial control over decomposition, such as MOMOS (Pansu *et al.*, 2010; Treseder *et al*., 2011; Todd-Brown *et al*., 2012; Pansu *et al.*, 2014). The biogeochemical processes simulated by the DNDC were generally based on first-order kinetics for decomposition and Michaelis-Menten kinetics of two substrates for nitrification and denitrification, which only the

parameterized growth and death of nitrifiers and denitrifiers were considered (Li, 2000). However, due to the global application and validation of DNDC (e.g., Chen *et al*., 2008; Giltrap *et al*., 2010; Cui *et al*., 2014, Zhang *et al*., 2015), the biogeochemical processes of DNDC were selected in the CNMM-DNDC despite some deficiencies in simulating microbial biomass." (**See lines 121-128 in the revised manuscript**). "Compared with the empirical model, one key advantage of the process-oriented models is that the models are independent of the local parameterization (Zhang *et al.*, 2015). In this study, default internal parameter combinations of biogeochemical processes were used for the original and modified models, which have been applied in the catchment simulation in the subtropical region (Zhang *et al.*, 2018b), due to the limited field observations (only one year) for both calibration and validation. The biogeochemical processes were predicted by the first-order and Michaelis-Menten kinetics in the CNMM-DNDC based on some defined parameters of flow fractionation. For instance, there are 17 parameters related with $N_2O$ emission in the module of denitrification (Table S6), which would inevitably increase the uncertainty of simulation. Houska *et al.* (2017) found that hydro-biogeochemical models can be right for the wrong reasons, such as matching greenhouse gas emissions while failing to simulate soil moisture, which emphasized the importance for simultaneous validations of multi-variables. Thus, simultaneous validations of $CH_4$ and $N_2O$ fluxes, as well as soil environment variables, were necessary for comprehensive evaluation of the model performance. In addition, the microbial ecology was recently recommended to be integrated into the biogeochemical model using a smaller number of well-defined kinetic parameters, such as MOMOS (Pansu *et al.*, 2010; Treseder *et al*., 2011). Therefore, direct control of microbial on biogeochemical processes, such as the stoichiometry of decomposer, is required to be included in the CNMM-DNDC in near future." (**See lines 397-410 in the revised manuscript**).

Also the article needs to be improved by more explanations about the CNMM part.
**Revised**.
The description about the CNMM has been added as the reviewer suggested.
"The soil moisture was calculated based on the mass balance of precipitation, irrigation, evapotranspiration, vertical flow, lateral flow and water from a rising water table. The total water that can be infiltrated during each time step was determined by a defined maximum infiltration rate. Darcy's law was applied for predicting the vertical water flow in the soil profile. A cell-by-cell approach using a kinematic approximation was applied to route the saturated overland and subsurface flow based on DEM. The stream flow was estimated using a cascade of linear channel reservoirs (Wigmosta *et al.*, 1994). For plant growth, gross primary production was simulated using Farquhar *et al.* (1980) for $C_3$ and Collatz *et al.* (1992) for $C_4$, with annual primary productivity calculated as the residue of gross primary production and autotrophic respiration." (**See lines 130-137 in the revised manuscript**).

Some surprising choices must be better justified in the text, like why to predict temperature in all the profile for two systems when moisture is described in the 3 systems but only in the surface layer. The particular remarks below must be taken into account.

Revised.

The data used for model validation is determined by the field observations. Due to the limitation of field observations, only the simulated topsoil temperature for the alpine forest and topsoil moisture for the three alpine ecosystems were able to be validated by observations (See lines 244-245 and 254-255 in the revised manuscript).

Particular remarks

L37 Is this 1st sentence a little banal?

Revised.

The sentence has been revised.

"During the last 50 years, the extraordinary changes in the nitrogen and carbon cycles have occurred globally, which are essential components of ecosystems (e.g., Galloway *et al.*, 2008; Canfield *et al.*, 2010)." (See lines 36-37 in the revised manuscript).

L 60-62 the listed biogeochemical models is not exhaustive and limited to relatively old models sometimes subjected to critics (over parameterization, functional role of microorganisms...), e.g. the model MOMOS is ignored, please complete the list in such a way to better situate your work in the literature data.

Revised.

The sentences have been revised (See lines 60-65 in the revised manuscript).

2.1 paragraphs: the original paper on DNDC includes a sub model for denitrification but not for methane emission; please clarify this part.

Revised.

In DNDC, the sub module of fermentation is closely related with methane emission, which includes the process of methane production, oxidation and transportation. Thus, the reference of Li (2000, 2007, 2016) has been added (See lines 144-145 in the revised manuscript).

Li, C., 2000. Modeling trace gas emissions from agricutural ecosystems. Nutr. Cycl. Agroecosyst. 58, 259–276.

Li, C., 2007. Quantifying greenhouse gas emissions from soils: scientific basis and modeling approach. Soil Sci. Plant Nutr. 53, 344–352.

Li, C., 2016. Biogeochemistry: Scientific Fundamentals and Modelling Approach. Tsinghua University Press, Beijing. Pp. 530. (In Chinese)

The only denitrification submodel includes 17 parameters which must be combined to the numerous parameters regulating various splitting's in the DNDC core! By two points you can adjust a right line, but also a parabola, then more and more sophisticated functions by increasing the number of parameters, and the fittings will be already OK when evaluated by the statistic laws. All the models will predict the two points. You must discuss about the number of parameters in the paper (Ockham razor)

**Revised**.

We fully agreed with the reviewer. The related discussion has been added as the reviewer suggested.

"Compared with the empirical model, one key advantage of the process-oriented models is that the models are independent of the local parameterization (Zhang *et al.*, 2015). In this study, default internal parameter combinations of biogeochemical processes were used for the original and modified models, which have been applied in the catchment simulation in the subtropical region (Zhang *et al.*, 2018b), due to the limited field observations (only one year) for both calibration and validation. The biogeochemical processes were predicted by the first-order and Michaelis-Menten kinetics in the CNMM-DNDC based on some defined parameters of flow fractionation. For instance, there are 17 parameters related with $N_2O$ emission in the module of denitrification (Table S6), which would inevitably increase the uncertainty of simulation. Houska *et al.* (2017) found that hydro-biogeochemical models can be right for the wrong reasons, such as matching greenhouse gas emissions while failing to simulate soil moisture, which emphasized the importance for simultaneous validations of multi-variables. Thus, simultaneous validations of $CH_4$ and $N_2O$ fluxes, as well as soil environment variables, were necessary for comprehensive evaluation of the model performance. In addition, the microbial ecology was recently recommended to be integrated into the biogeochemical model using a smaller number of well-defined kinetic parameters, such as MOMOS (Pansu *et al.*, 2010; Treseder *et al.*, 2011). Therefore, direct control of microbial on biogeochemical processes, such as the stoichiometry of decomposer, is required to be included in the CNMM-DNDC in near future." (**See lines 397-410 in the revised manuscript**).

L111-112 OK for hydro-geochemical but there is a doubt for biogeochemical, please discuss more deeply your choices considering all the literature data. The same for biogeochemistry in line 112; in contrast, paragraph 2 of 2.1.1 explain a particular interest of CNMM-DNDC

**Revised**.

More detailed descriptions have been added with updated references.

"For the new generation of biogeochemical models, the microbial ecology was integrated into the biogeochemical models, which represents direct microbial control over decomposition, such as MOMOS (Pansu *et al.*, 2010; Treseder *et al*., 2011; Todd-Brown *et al*., 2012; Pansu *et al.*, 2014). The biogeochemical processes

simulated by the DNDC were generally based on first-order kinetics for decomposition and Michaelis-Menten kinetics of two substrates for nitrification and denitrification, which only the parameterized growth and death of nitrifiers and denitrifiers were considered (Li, 2000). However, due to the global application and validation of DNDC (e.g., Chen *et al*., 2008; Giltrap *et al*., 2010; Cui *et al*., 2014, Zhang *et al*., 2015), the biogeochemical processes of DNDC were selected in the CNMM-DNDC despite some deficiencies in simulating microbial biomass." (**See lines 121-128 in the revised manuscript**).

Eq5: It looks like a sum, not a weighted average? Should j be defined in L148?
**Revised**.
The sentence and Eq. 5 have been revised to make it clear.
"The dynamic soil heat capacity ($C_l$, J m$^{-3}$ ℃$^{-1}$) is the weighted average of the heat capacity for five constituents, including organic matter ($C_{l, OM}$), minerals ($C_{l,Min}$), water ($C_{l, Water}$), ice ($C_{l, Ice}$) and air ($C_{l, Air}$) (Eq. 5). The values of heat capacity for organic matter, minerals, water, ice and air were $2.5 \times 10^6$, $2.0 \times 10^6$, $4.2 \times 10^6$, $2.1 \times 10^6$ and $1.2 \times 10^3$ J m$^{-3}$ ℃$^{-1}$, respectively (Huang, 2000)." (**See lines 170-173 and Eq. 5 in the revised manuscript**).

Eq.6: product or geometric mean?
**Revised**.
The detailed the calculation steps of thermal conductivity ($k_l$) have been added as the reviewer suggested. (**See Eq. 6–13 in the revised manuscript**).

L148-151: is there references data for the values of C and k for each constituent j?
**Revised**.
The references have been added as the reviewer suggested (**See lines 172-173 and 176-177 in the revised manuscript**).

L168: reference for this opaque chamber method? Is not a risk of perturbation by elimination of solar radiation?
**Revised**.
The reference has been added with the explanation for effects of the solar radiation.
"...using the gas chromatograph-based static opaque chamber method (Zhang *et al.*, 2018a) at three sites ... Each chamber was wrapped with a layer of styrofoam and aluminium foil to mitigate temperature increases inside the enclosures due to the heating of solar radiation." (**See lines 196-197 and 199-201 in the revised manuscript**).

L189-190: could the layer description be clarified?

**Revised**.

The sentence has been revised to make it clear.

"The layer thicknesses of the soil (0−1.5 m) were 1, 5 10 and 50 cm for the depth of 0–10, 10–20, 20–100 and 100–150 cm, respectively. The layer thicknesses of the bedrock (1.5−35 m) were 3.5 and 31m for the depth of 1.5−4.0 and 4.0−35 m, respectively." (**See lines 222-224 in the revised manuscript**).

Table1: Could the legend remember the meaning of IA, NSI and R2? I suppose P is the probability of rejection, but for what test? Are the P columns necessary since P is always <0.001?

**Revised**.

The detailed information about the statistics has been added in Table 1 (**See Table 1 in the revised manuscript**).

Fig.1: legend in 1a is not very clear, perhaps write "observed" in red color; the same for other Figs

**Revised**.

The legends of all figures have been revised as the reviewer suggested (**See Figs. 1−4 in the revised manuscript**).

L210: ZIR does not appear in table 1

**Revised**.

The title line of Table 1 has been revised to detail the slope, $R^2$ and P of ZIR. The equation applied for the zero-intercept univariate linear regression (ZIR) of the observations against the simulations has been added in Table S5. (**See Table 1 in the revised manuscript**).

L216: The range values does not correspond to that of Table 1, please clarify

**Revised**.

The inconsistent range values has been checked and revised (**See lines 255-256 in the revised manuscript**).

L218: where is Fig S2? (the same for all S figures, I suppose supplementary not visible for me)

**Revised**.

The supplementary materials have been uploaded with the revised manuscript (**See the supplementary materials after the revised manuscript**).

Improve coherencies in Figs or more explain your choices: Fig 1 shows soil temperature for two systems at all soil layers; in contrast, Fig.2 shows moisture in the 3 systems but only in the 0-6 cm layer

**Revised**.

The data used for model validation is determined by the field observation. Due to the limitation of field observations, only the simulated topsoil temperature for the alpine forest and topsoil moisture for the three alpine ecosystems were able to be validated by observations (**See lines 244-245 and 254-255 in the revised manuscript**).

Fig3: format date of x axis not very clear

**Revised**.

The formats of date for all figures have been revised to make them clear (**See Figs. 1−4 in the revised manuscript**).

L223: is the term "Water movement" exaggerated when you speak only of the surface water?

**Revised**.

The sentence has been revised as the reviewer suggested (**See lines 262-263 in the revised manuscript**).

Fig.6:y axis legend not clear

**Revised**.

The y axis of Fig. 4 has been revised to make them clear (**See Fig. 4 in the revised manuscript**).

L293-294: should you present succinctly this concept of the CH4 balloon in material and methods?

**Revised**.

The contents have been added as the reviewer suggested.
 "Methane production and oxidation occurred simultaneously and were determined by the sizes of the aerobic (production) and anaerobic (oxidation) microsites, which were defined by an Eh calculator in terms of an "anaerobic balloon" ("$CH_4$ balloon") (Li, 2007)." (**See lines 145-147 in the revised manuscript**).

L351: define TP

**Revised**.

The descriptions have been revised throughout the manuscript (**See lines 25, 90-91 103, 187 and 420 in the revised manuscript**).

---

## Author Response (AR2)

As you will see the feedbacks from referees confirm your paper is close to be accpeted for publication. Nevertheless they rise a number of comments that need to be addressed before publication.

**Revised**.

The related contents have been revised as the reviewer suggested.

Another important point concerns the free accessibility of scientific ressources, a policy of the Journal. I would ask you to upload your code and data on a public available repository, instead of this standard data availabity statement "The model, input and output databases can be obtained from the first author and all the observed data sets used in this study can be available from the co-authors." Such statements hinder not only the review process, but also the reproducability of the scientific work.

**Revised**.

The model, input, output and code can be obtained from http://doi.org/10.6084/m9.figshare.14685441 (**See line 426 in the revised manuscript**).

I would like to thank the authors for their throughout revision, which improved the manuscript signifcantly. Before potential publication, I have just some minor revision points:

Thanks for you evaluation.

Line 233: Please stay consistent and use the unit "meter" to describe the soil layer thickness.

**Revised**.

The unit has been revised to stay consistent (**See lines 222-223 in the revised manuscript**).

Table 1: For the reader it would be easier, if the better performing values comparing original and modified model would be bold for each of the objective functions. ZIR-P is not introduced in the Materials und Methods nor in the Table captions.

**Revised**.

The table has been revised with the explanations for the ZIR-P. (**See Table 1 and lines 237-238 in the revised manuscript**).

Line 318 -323: I still think this scenario/experiment does not fit in the manuscript. In this revised version, it is not properly introduced. Neither in the introduction, material and methods and not much discussed in the discussion chapter. Further, it has not much to do with the rest of the paper. I suggest the authors to remove this and the corresponding parts (e.g. Line 243-244) and see this as a minor revision. Deleting this part is not meant to discourage the authors. I think the idea of such a scenario analysis is interesting and I recommend the authors to publish the idea properly worked out

and present it in an additional publication.

**Revised**.

The related contents have been deleted as the reviewer suggested.